

# Anatomy of the eigenstates distribution:
# A quest for a genuine multifractality

Anton Kutlin[1,2]* and Ivan M. Khaymovich[3,4]

**1** Abdus Salam International Center for Theoretical Physics,
Strada Costiera 11, 34151 Trieste, Italy
**2** Max Planck Institute for the Physics of Complex Systems,
Nöthnitzer Straße 38, 01187-Dresden, Germany
**3** Nordita, Stockholm University and KTH Royal Institute of Technology,
Hannes Alfvéns väg 12, SE-106 91 Stockholm, Sweden
**4** Institute for Physics of Microstructures, Russian Academy of Sciences,
603950 Nizhny Novgorod, GSP-105, Russia

* anton.kutlin@gmail.com

## Abstract

Motivated by a series of recent works, an interest in multifractal phases has risen as they are believed to be present in the Many-Body Localized (MBL) phase and are of high demand in quantum annealing and machine learning. Inspired by the success of the Rosenzweig-Porter (RP) model with Gaussian-distributed hopping elements, several RP-like ensembles with the fat-tailed distributed hopping terms have been proposed, with claims that they host the desired multifractal phase. In the present work, we develop a general (graphical) approach allowing a self-consistent analytical calculation of fractal dimensions for a generic RP model and investigate what features of the RP Hamiltonians can be responsible for the multifractal phase emergence. We conclude that the only feature contributing to a genuine multifractality is the on-site energies' distribution, meaning that no random matrix model with a statistically homogeneous distribution of diagonal disorder and uncorrelated off-diagonal terms can host a multifractal phase.

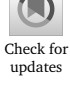

# 1 Introduction

A study of eigenstates of random Hamiltonians is crucial for understanding the spectral and transport properties of various systems. For example, basis-invariant ensembles like the Gaussian Orthogonal Ensemble (GOE) or Gaussian Unitary Ensemble (GUE) [1] have basis-invariant distributions of eigenstates, meaning that, on average, all their components are of the same order. This leads to the ergodicity of eigenstates and, thus, makes the system conducting. On the other hand, the 1d Anderson model's eigenstates are exponentially localized around their maxima, meaning that any local perturbation to the system only affects a finite number of such eigenstates, and the system is an insulator.

The examples of ergodic and localized states are just the opposite ends of the broad spectrum of what is available. For example, systems at the critical points of the Anderson transition between the ergodic and localized phases are known to host the so-called fractal, or even multifractal, eigenstates [2]. In a thermodynamic limit of the system size going to infinity, such critical eigenstates occupy an infinite number of sites, being still a measure zero of the total system size. The difference between fractal and multifractal eigenstates is even more subtle than that between fractal and localized ones. This is the difference in the probability density functions (PDF) of the eigenstate coefficients in these two cases. Fractal states are charac-

terized by a single power-law tail in PDF which usually provides the only energy/time scale in the system in addition to the standard ones: the bandwidth and the mean level spacing. Here, by an additional energy/time scale, we mean the Thouless energy/time, like in [3, 4], which appears to be the only parameter, needed to collapse the power-law distributions of wave-function coefficients. Multifractal states, in turn, have multiple (running) power-law exponents at different scales. This means that the above distribution of eigenstate coefficients cannot be approximated by a single power-law, but needs many power-law-like pieces at different wave-function amplitudes. The simplest example of such a multifractal distribution would be a log-normal one, known to be the signature of weak multifractality [2]. In this respect, the multifractal distribution, being the most general type of smooth distribution, provides the set of energy scales and, thus, such systems usually have more rich time evolution and hierarchical local spectral structure, needed for the quantum-algorithm speed-up applications [5] and faster learning of artificial neural networks [6]. For these applications, the robust multifractal phases of matter are of crucial importance.

As another motivation, we consider a series of recent works [7–10], providing numerical evidence of the Hilbert space multifractality in the entire phase of many-body localization (MBL) in disordered interacting quantum systems. Since the direct study of MBL is challenging numerically and analytically, it is of particular interest and high demand to develop some proxy models that mimic the necessary properties of the MBL systems but are easier to tackle. One of such proxies is an Anderson model on random regular graphs (RRG) [11, 12], possessing a hierarchical structure, similar to the many-body Hilbert space. It was believed to host multifractal states [13–21], even before similar claims about the actual many-body systems have been made [12]. To simplify the problem even more, a random-matrix proxy to the RRG was proposed, namely the log-normal Rosenzweig-Porter model (LN-RP) [18, 22]. Due to the multifractal and heavy-tailed nature of the log-normal distribution, it was believed to host a genuine multifractal phase. However, the suggested mapping implied that the RRG corresponds to the such parameters of the LN-RP models, where the multifractal phase disappears at the tricritical Anderson transition point, meaning that the observed RRG multifractality can be just a finite-size effect, see also [23–28].

Nevertheless, the very existence of the multifractal phase in the log-normal Rosenzweig-Porter model (unlike the fractal one in the Gaussian RP [29]) has never been proven mathematically and was based on the numerical evidence and the heuristic argument predicting multifractality for any RP model with heavy-tailed distribution, e.g., a Lévy-RP model [30, 31]. Given that the Gaussian Rosenzweig-Porter model [32–37], eight years after the discovery of the fractal phase there [38], is still the only analytically tractable model [29, 30, 39, 40] hosting an entire phase of fractal states, it makes sense to look for an analytical approach to the other RP models, like Lévy- [30, 31] or LN-RP [3, 18, 22], as such attempt does not look hopeless. In this paper, we show that it is indeed possible.

The paper is organized as follows. In Sec. 2, we define our main object of study, the spectrum of fractal dimensions, and discuss how it allows us to distinguish the multifractality from the fractality. Section 3 introduces a graphical approach to deal with the spectra of fractal dimensions (SFDs) of different independent random variables. Methodologically, this section contains the paper's main results. In Sec. 4, we demonstrate the application of the developed machinery to the Gaussian Rosenzweig-Porter model and compare our result to the previously known ones. Section 5 shows a predictive power of the above-developed method by applying it to the Lévy-RP model and demonstrates that its local density of states (LDOS) has a fractal, but not multifractal distribution. In Sec. 6, we do the same for the log-normal RP model and, again, claim the absence of LDOS multifractality. In Sec. 7, we analyze if the lack of LDOS multifractality implies the absence of eigenstates multifractality and conclude that, for models with RP-like LDOS SFDs, it does. Finally, in Sec. 8, we prove that no RP-like model,

i.e., a model with a regular on-site disorder, no correlations between hopping elements, and no spatial structure, can host a multifractal phase.

## 2  Multifractality and the spectrum of fractal dimensions

For a non-negative random variable $X$ (e.g., the eigenstate amplitude $|\psi(i)|^2$ at site $i$), the spectrum of fractal dimensions (SFD) $f_X(\alpha; N)$ parameterizes the probability density function $p_X(x)$ as

$$p_X(x)\mathrm{d}x = p_X(N^{-\alpha})\ln(N)N^{-\alpha}\mathrm{d}\alpha = \ln(N)N^{f_X(\alpha;N)-1}\mathrm{d}\alpha. \tag{1}$$

For the wave-function amplitudes, $N$ is the system size, which is considered to be large. In other words, we focus mainly on the large-$N$ limit $f_X(\alpha) = \lim_{N\to\infty} f_X(\alpha; N)$ if it is not stated explicitly otherwise. The intuition behind the spectrum of fractal dimensions is as follows: in the set of $N$ independent samples of $X$, we will find around $N^{f_X(\alpha;N)}$ samples of the order of $X \sim N^{-\alpha}$. Note that the parameter $N$ can, in principle, be any number, but, in our physical applications, we will associate it with the system size.

The SFD contains all the necessary information to extract the eigenstate fractal dimensions $D_q$ and the critical exponents $\tau_q = (q-1)D_q$ [2]. In the large-$N$ limit, $\tau_q$ is given by the Legendre transform of $f(\alpha)$ in the saddle-point approximation:

$$\tau_q \stackrel{\mathrm{def}}{=} -\log_N \mathrm{IPR}_q \simeq -\log_N\left[N\left\langle|\psi(i)|^{2q}\right\rangle\right] \sim -\log_N\left(\int \mathrm{d}\alpha N^{-\alpha q}N^{f(\alpha)}\right) \xrightarrow[N\to\infty]{} q\alpha_q - f(\alpha_q). \tag{2}$$

Here the first equality is the definition of $\tau_q$ via the logarithm $\log_N$ of base $N$ of the inverse participation ratio $\mathrm{IPR}_q = \sum_i |\psi(i)|^{2q}$, with the sum approximated by the averaging over the probability distribution (1), noted by $\langle\ldots\rangle$. In the r.h.s. of (2), the Legendre transform is given by the parameter $\alpha_q$, minimizing $q\alpha - f(\alpha)$. For a smooth $f(\alpha)$, $\alpha_q$ is defined as the solution of the equation $f'(\alpha_q) = q$.[1] A pictorial representation of this minimization is shown in Fig. 1.

From this result, one can readily identify at least two important values of $\alpha$: $\alpha_0$ and $\alpha_1$. The value $\alpha_0$ corresponds to the maximum of the SFD, $f(\alpha_0) = \max_\alpha\{f(\alpha)\} = 1$.[2] Using the intuitive meaning of SFD mentioned above, one can deduce that, in the sufficiently large sample, the realizations with $x \sim N^{-\alpha_0}$ will prevail over any other values of $x$, making this value *typical* for the ensemble. On the other hand, the *mean* value of $x$ corresponds to $\alpha_1$. For example, the SFDs of normalized wave functions with $1 = \sum_{i=1}^N |\psi(i)|^2 = N\left\langle|\psi(i)|^2\right\rangle$ always lie below the line $f(\alpha) = \alpha$ and have at least one common point with this line in the thermodynamic limit. Finally, this interpretation allows us to give a mathematically strict definition of the support set dimension $D$ [41]. To see this, let us calculate $D_1$ as a limit of $D_q$ for $q \to 1$ via the L'Hôpital's rule:

$$D_1 = \lim_{q\to 1} D_q = -\sum_i |\psi(i)|^2 \log_N |\psi(i)|^2 \sim \int \alpha N^{f(\alpha;N)-\alpha}\ln N \mathrm{d}\alpha \xrightarrow[N\to\infty]{} \alpha_1. \tag{3}$$

Thus, since $\alpha_1$ is responsible for the normalization, $D_1 = \alpha_1$ represents the actual, coherent with its physical meaning, support set dimension.

As it has been just shown, in a generic case, the different fractal dimensions are given by the different points of the SFD. However, the SFD can also have discontinuities in its first derivative. In this case, each $\alpha$, corresponding to one of such discontinuities, contributes to $D_q$

---

[1]If the relations provides several solutions for $\alpha_q$, one should pick the one maximizing $f(\alpha_q) - \alpha_q$.

[2]The maximum value of $f(\alpha)$ is always 1 due to the normalization condition of the probability density function (1).

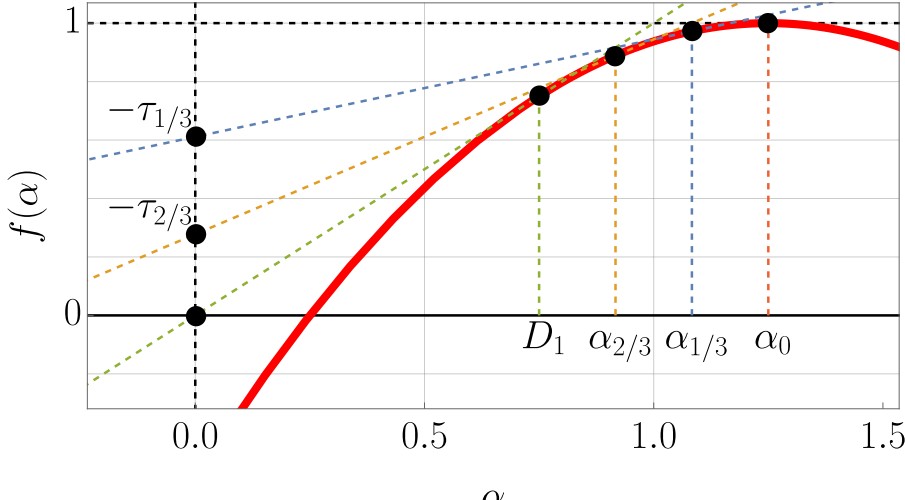

Figure 1: Graphical representation of the relation between the spectrum of fractal dimensions $f(\alpha)$ and the critical exponents $\tau_q$, Eq. (2). Here, the solid red line shows the SFD of a certain distribution, while the tangent dashed lines of different colors with the slopes 1 (green), 2/3 (orange), and 1/3 (blue) correspond to Legendre transform from the r.h.s of (2). The label "$D_1$" corresponds to the value of $\alpha = \alpha_{q\to 1}$, responsible for the normalization $N\langle N^{-\alpha}\rangle \propto N^0$ and, hence, represents the actual, correctly defined, support set dimension [41], see the main text for more details.

in an entire range of different $q$'s, meaning that the same fraction of the eigenstate components contributes to different fractal dimensions and $D_q$ stays constant in the corresponding range of $q$'s. The eigenstates are called multifractal [2] if $D_q$ is *not* constant for integer positive $q$ or, in other words, it $f(\alpha)$ *does* have finite values at $\alpha < \alpha_1$.[3] Otherwise, we will talk about fractality.

Note that the concept considered here is very similar in spirit to the large-deviation theory (see, e.g., the review [42]), where all the physical objects, like the partition or generating functions, probability distributions of extensive variables are considered in the same way. The main difference is in the large scaling parameter: in the large-deviation theory it is usually time, while in our case it is given by the logarithm of the matrix size $\ln N$. For more examples of such analogies, one can also look into [43] and references therein.

## 3 Graphical algebra

Working with probabilities, we often need to calculate probability distributions of composite random variables, i.e., a PDF of a sum or a product. Such quantities can be obtained via proper integral convolutions or using characteristic functions. However, the calculation difficulty grows rapidly with the complexity of the composite random variable's expression. Working with fractal spectra brings a whole new life to such operations due to the possibility of approximate integration using the Laplace method. In this section, we derive simple pictorial rules allowing the calculation of composite random variables' SFDs on the fly.

Below, we will consider the following operations: an exponentiation of a random variable (Sec. 3.1), a sum of two independent random variables (i.r.v.s) (Sec. 3.2), an "ensemble mix-

---

[3]Here and further we define $\alpha_1$ as the leftmost of all equivalent ones if $f(\alpha)$ has a straight segment of a finite length with the slope 1.

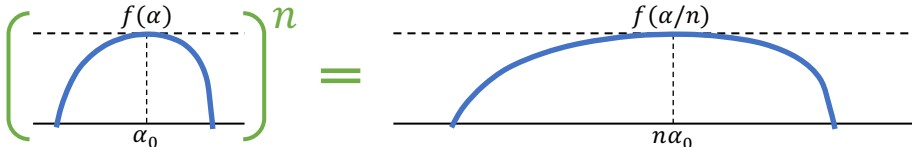

Figure 2: Pictorial representation of random variable exponentiation. In this case, $n$ is assumed to be greater than one. Here and below, we omit labeling of the axis, always assuming the horizontal axis to represent $\alpha$ and the vertical axis to represent $f(\alpha)$. The dashed horizontal line marks the level $f(\alpha) = 1$, the solid horizontal line corresponds to $f(\alpha) = 0$, and the labels "$\alpha_0$" and "$n\alpha_0$" correspond to the typical values of $\alpha$ before and after the exponentiation.

ture" of several independent random variables with different distributions (Sec. 3.3), and a product of two i.r.v.s (Sec. 3.4). For the exponentiation and the ensemble mixture, the derivations of the graphical rules are simple regardless of the random-variable (r.v.) distributions. The derivations for the rest two operations, become more straightforward under the assumption that their SFDs are convex functions. However, this does not limit the applicability of the results since sums and products of any two i.r.v.s can always be decomposed into a superposition of the operations involving i.r.v.s with convex SFDs only.

## 3.1 Raising a random variable to a power

We start by considering the simplest case, namely, by expressing a spectrum of fractal dimensions $f_{X^n}(\alpha)$ of $X^n$ in terms of the spectrum of fractal dimensions $f_X(\alpha)$ of $X$. Since, by definition $p_X(N^{-\alpha}) \propto N^{f_X(\alpha)+\alpha-1}$ and, by variables substitute, $p_{X^n}(x) = n^{-1}x^{1/n-1}p_X(x^{1/n})$, we get that

$$p_{X^n}(N^{-\alpha}) = N^{f_{X^n}(\alpha)+\alpha-1} \propto N^{(1-1/n)\alpha}N^{f_X(\alpha/n)+\alpha/n-1}, \tag{4}$$

and, hence,

$$f_{X^n}(\alpha) = f_X(\alpha/n). \tag{5}$$

Pictorially, this is just an $n$-times stretching of the spectrum of fractal dimensions in a horizontal direction with a fixed point $\alpha = 0$, see Fig. 2. For the negative $n$, this is also a reflection with respect to the vertical axis.

## 3.2 Sum of two independent random variables

The main point for the sum rule, written in terms of SFDs, is that the expression $x + y = N^{-\alpha_X} + N^{-\alpha_Y}$ is binary: it equals $N^{-\alpha_X}$ if $\alpha_Y > \alpha_X$ and $N^{-\alpha_Y}$ otherwise. In other words, as soon as $x < y$, we can always neglect $x$ completely, and vice versa. If $x + y = N^{-\alpha}$, then $\alpha = \min\{\alpha_X, \alpha_Y\}$. It allows us to write that

$$N^{f_{X+Y}(\alpha)} \propto N^{f_X(\alpha)}P(\alpha_X < \alpha_Y|\alpha_X = \alpha) + N^{f_Y(\alpha)}P(\alpha_Y < \alpha_X|\alpha_Y = \alpha). \tag{6}$$

Without loss of generality, let us assume that $\alpha_0(X) < \alpha_0(Y)$, i.e. that typically $Y$ is negligible. Then there are three cases: $\alpha < \alpha_0(X)$, $\alpha_0(X) < \alpha < \alpha_0(Y)$, and $\alpha > \alpha_0(Y)$. let us consider them one by one.

The case $\alpha < \alpha_0(X)$ is simple: since $N^{-\alpha}$ is bigger than the typical values of both $X$ and $Y$,

$$N^{f_{X+Y}(\alpha)} \propto N^{f_X(\alpha)} \cdot 1 + N^{f_Y(\alpha)} \cdot 1 \implies f_{X+Y}(\alpha) = max\{f_X(\alpha), f_Y(\alpha)\}, \tag{7}$$

here, both $P(\alpha_X < \alpha_Y|\alpha_X = \alpha)$ and $P(\alpha_Y < \alpha_X|\alpha_Y = \alpha)$ are of the order of unity as the typical values $\alpha_0(X)$, $\alpha_0(Y)$ are in the interval $\alpha_{X,Y} > \alpha$.

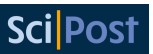

Figure 3: Pictorial representation of a sum of two i.r.v.s with convex SFDs.

The case $\alpha > \alpha_0(Y) > \alpha_0(X)$ is not much different: since now the typical values of $X$ and $Y$ are both bigger than $N^{-\alpha}$,

$$N^{f_{X+Y}(\alpha)} \propto N^{f_X(\alpha)}N^{f_Y(\alpha)-1} + N^{f_Y(\alpha)}N^{f_X(\alpha)-1} \implies f_{X+Y}(\alpha) = f_X(\alpha) + f_Y(\alpha) - 1, \quad (8)$$

here, the corresponding conditional probabilities are denoted[4] by $N^{f_{X,Y}(\alpha)-1}$. And, as both $f_X(\alpha)$ and $f_Y(\alpha)$ in this region are smaller than one, this result shows a suppression of zeros which is quite logical: the more positive terms we add, the fewer probable we get something small.

Finally, in the intermediate case, $\alpha_0(X) < \alpha < \alpha_0(Y)$, the probability $P(\alpha_X < \alpha_Y | \alpha_X = \alpha)$ is of the order of one, while the probability $P(\alpha_Y < \alpha_X | \alpha_Y = \alpha)$ is of the order of $N^{f_X(\alpha)-1}$, hence,

$$N^{f_{X+Y}(\alpha)} \propto N^{f_X(\alpha)} + N^{f_Y(\alpha)}N^{f_X(\alpha)-1} \implies f_{X+Y}(\alpha) = f_X(\alpha). \quad (9)$$

In the last equality we used the fact that $f_Y(\alpha) - 1 < 0$ at $\alpha < \alpha_0(Y)$. A pictorial representation of this result is shown in Fig. 3.

### 3.3 Weighted ensemble mixture of several independent random variables

Suppose that we are interested in the SFD of a random variable $X$ with a probability density function proportional to a weighted sum, $\sum_j w_j = 1$, of other probability density functions:

$$p_X(x) = \sum_i w_i p_{X_i}(x). \quad (10)$$

Such a definition corresponds to the notion of conditional probability and the chain rule. Indeed, going from the sum to an integral and making substitutions $w_i \to p(i)di$ and $p_{X_i}(x) \to p(x|i)$, we arrive at the probably more familiar expression

$$p_X(x) = \int p(x|i)p(i)di. \quad (11)$$

Now, the transformation of the PDF to the SFD in the saddle-point approximation gives the relation

$$f_X(\alpha) = \max_i \{f_i(\alpha) + f(i) - 1\}, \quad (12)$$

where $f(i)$ is the SFD corresponding to the probability density function $p(i)$. In the previous section, we already saw a similarly looking relation (7) for two discrete $i = X, Y$ values with the same weights. It was a particular case of this more general mix rule.[5]

This is the mix rule that allows us to consider explicitly only r.v.s with convex SFDs without losing generality, as it was mentioned earlier in the introduction to this chapter. Indeed, it is easy to notice that any SFD can be defined as a mix of convex SFDs, while both sum and product of two i.r.v.s, being bi-linear operations, can be represented as the mixes of sums and products of the convex SFDs.

---

[4]It is the consequence of the assumed convexity of the functions $f_{X,Y}(\alpha)$. Otherwise, we would have to write the probabilities as $\max_{\omega \geq \alpha} N^{f_{X,Y}(\omega)-1}$.

[5]The name 'mix rule' highlights the fact that $X$ can be seen as a random variable realizing in itself different ensembles with corresponding probabilities, i.e., a mixture of ensembles.

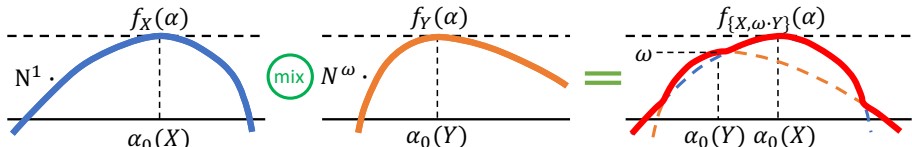

Figure 4: Graphical representation of the weighted mix rule. In this particular case, the resulting distribution consists of the 'blue' distribution with the weight $N^1$ and the 'orange' distribution with the weight $N^\omega$. Notice how the orange distribution in the r.h.s was shifted vertically according to its weight. After making such shifts for all involved SFDs, the resulting SFD is obtained by a simple envelope.

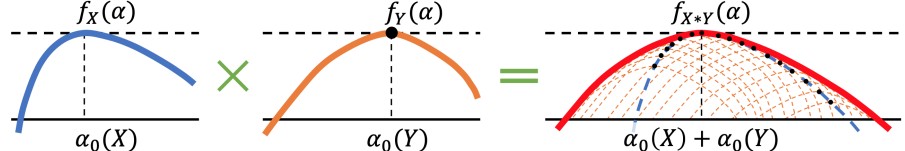

Figure 5: Graphical representation of a product of the two i.r.v.s in terms of the weighted mix. In this figure, the 'blue' SFD plays the role of weights, while the 'orange' SFD plays the role of the shifted distribution. Interchanging the roles of the two will not alter the final result. In the rightmost panel, there are many replications of the orange curve, shifted such that their maxima (marked by the black points) always lie on the blue curve. The envelope of this construction corresponds to the desired product.

## 3.4 Product of two independent random variables

To calculate the SFD of the product of two i.r.v.s $X \sim N^{-\xi}$ and $Y \sim N^{-\eta}$, let us consider the corresponding convolution in terms of $\xi$ and $\eta$:

$$N^{f_{XY}(\alpha)-1} \propto \int_{-\infty}^{\infty} \mathrm{d}\xi \mathrm{d}\eta\, \delta(\alpha-\xi-\eta) N^{f_X(\xi)-1} N^{f_Y(\eta)-1} = \int_{-\infty}^{\infty} \mathrm{d}\xi\, N^{f_X(\xi)+f_Y(\alpha-\xi)-2}. \tag{13}$$

Calculating this integral in the saddle-point approximation, we arrive at the expression suspiciously similar to the one from the mix rule:

$$f_{XY}(\alpha) = \max_{\xi}\{f_X(\xi) + f_Y(\alpha-\xi) - 1\}. \tag{14}$$

And this is not a surprise: the product can be understood purely in terms of the previously introduced mix rule (Sec. 3.3). To see this, notice that multiplication by a *constant* $N^{-s}$ results in a horizontal shift of the multiplied SFD by the distance $s$. Thus, the multiplication by an arbitrarily distributed random variable $X = N^{-\alpha}$ can be now considered as a superposition of different horizontal shifts $\alpha$ (multiplication by a constant) with different vertical shifts $f(\alpha)-1$ (the constant's weights), restoring the same mix rule, see Fig. 5.

## 3.5 Extensive sums of i.i.d random variables

In Sec. 3.2, we have already seen how to calculate a spectrum of fractal dimensions of a sum of two independent random variables. It can be easily generalized to a sum of any finite number of r.v.s. At the same time, we can imagine that an *extensive* sum of r.v.s has to have something to do with its typical value, which does not follow from the finite sum rule, e.g., a central limit theorem predicts a square-root dependence of the typical value with the number of terms in

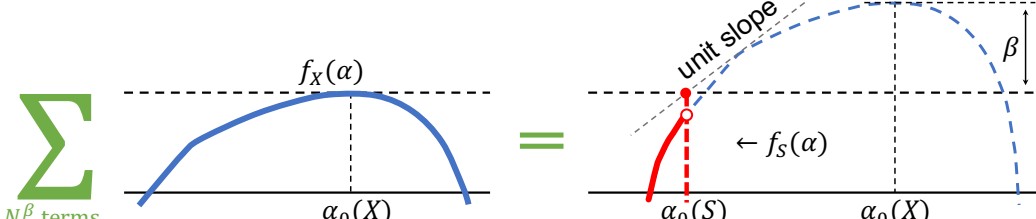

Figure 6: Graphical representation of the generalized central limit theorem. The summation result is shown in red.

the sum. To make the required generalizations, let us consider a non-negative random variable $X$ defined by its spectrum of fractal dimensions $f_X(\alpha)$ and a random variable $S$ defined as a sum of $N^\beta$ independent copies of the random variable $X$:

$$s = \sum_{i=1}^{N^\beta} x_i \,. \tag{15}$$

Our task now is to calculate $f_S(\alpha)$.

We start by focusing on such values of $x = N^{-\alpha}$ that appear in any typical sample of size $N^\beta$. Specifically, let us consider $\alpha$ such that its typical number of realizations $N^\beta p_{\alpha(X)}(\alpha) \propto N^{f_X(\alpha)-1+\beta}$ is large enough, i.e., $\alpha \in \Omega = \{\alpha | f_X(\alpha) - 1 + \beta \geq 0\}$. Choosing $\alpha_i \in \Omega$, one can estimate a probability to find exactly $N^{g_i-1+\beta}\delta\alpha$ terms of the order of $N^{-\alpha_i}$ in a particular realization of the sum as $\exp\{-\left(N^{g_i} - N^{f_X(\alpha_i)}\right)^2 / 2N^{f_X(\alpha_i)+1-\beta}\delta\alpha\}$ which is double-exponentially small in $g_i$, provided $g_i \neq f_X(\alpha_i)$. Thus, neglecting the double-exponentially small contributions, one can write a typical value of the sum as [22, 44]

$$s_{typ} \propto \int_\Omega d\alpha N^{-\alpha} N^{f_X(\alpha)-1+\beta} \propto N^{-\min_\Omega\{\alpha-f_X(\alpha)+1-\beta\}} \,, \tag{16}$$

meaning that $\alpha_0(S) = \min_\Omega\{\alpha - f_X(\alpha) + 1 - \beta\}$. In addition, one can deduce that $f_S(\alpha > \alpha_0(S)) = -\infty$, which is just a consequence of the 'zeros-suppression' effect mentioned in Sec. 3.2 in its extreme case.

At the same time, the values of $\alpha$ corresponding to $f_X(\alpha) + \beta < 1$ with $\alpha < \alpha_0(S)$ are unlikely to be represented in a typical realization of the sum even by a single term. However, in case of such an unlikely event, the whole sum will be determined by this very contribution. Thus, these rare events should be handled with the mix rule (Sec. 3.3): a probability density for such an event to contribute equals to $N^\beta p_{\alpha(X)}(\alpha) \sim N^{f_X(\alpha_i)+\beta-1} \ll 1$, resulting in the following graphical rules for obtaining the SFD of an extensive sum:

1. Draw $f_X(\alpha) + \beta$.

2. Draw a unit-slope line having exactly one common point with $f_X(\alpha) + \beta$ in the area where $f_X(\alpha) + \beta \geq 1$.

3. A point $\alpha_0$ where this unit-slope line crosses the horizontal level $f = 1$ is our new typical value, according to (16). There is nothing to the right of this point due to the zeros-suppression effect.

4. To the left of this point $f_S(\alpha)$ just equals $f_X(\alpha) + \beta$.

As one can see from this construction, all tails with a slope greater than unity eventually die as $\beta \to \infty$, in agreement with the standard central limit theorem. In addition, in agreement with the generalized central limit theorem for the one-sided stable distributions, in order for distributions to be stable, the tails of their PDFs should decrease not faster than $\propto s^{-2}$, which is exactly what they do, see Sec. 5 for more details. For such distributions, the unit-slope line touches $f_X(\alpha) + \beta$ exactly at $f = 1$ for any $\beta > 0$, thus, $f_S(\alpha)$ never develops discontinuities, and the tail never dies.

We have just derived the rule for extensive summation in the graphical language. For the derivation of the same result in a more traditional mathematical fashion, see Appendix A.

## 4   Gaussian Rosenzweig-Porter model

The Gaussian Rosenzweig-Porter ensemble is essentially just an ensemble of $N \times N$ Gaussian orthogonal (or unitary) random matrices with broken rotational symmetry:

$$H_{RP} = H_0 + V, \qquad [H_0]_{ij} = \varepsilon_i \delta_{ij}, \, \varepsilon_i \in \mathcal{N}(0,1), \qquad V = N^{-\gamma/2} H_{GOE/GUE}, \qquad (17)$$

where the elements of $H_{GOE/GUE}$ are i.i.d. Gaussian r.v.s, with zero mean and unit variance. In this section, we show how the graphical rules defined above can help self-consistently calculate the spectrum of fractal dimensions of the local density of states of the model from the first principles.

We do it using the cavity method in its diagonal approximation [39, 45]. The idea of the cavity method is to use an exact expression relating diagonal elements of an $N \times N$ Green's function $G(z) = (z-H)^{-1}$ and an $(N-1) \times (N-1)$ reduced Green's function $G^{(i)}(z) = (z-H^{(i)})^{-1}$, where $H$ and $H^{(i)}$ differ by a single site $i$ ($H^{(i)}$ has a small "cavity"):

$$G_{ii}(z) = \left( z - \varepsilon_i - \sum_{j,k \neq i} V_{ij} G_{jk}^{(i)}(z) V_{ki} \right)^{-1}, \qquad (18)$$

here, $z = \varepsilon - i\eta$ is a complex-valued parameter with a small imaginary part $\eta$ to ensure the existence of $G(z)$. For $\eta > 0$, one can define a local density of states $\nu_i(\varepsilon)$ as

$$\nu_i(\varepsilon) = \frac{1}{\pi} \operatorname{Im} G_{ii}(z) = \sum_{n=1}^{N} \frac{\eta/\pi}{(\varepsilon - E_n)^2 + \eta^2} |\psi_n(i)|^2. \qquad (19)$$

Following [45, 46], we choose $\eta = N^\beta \delta_\varepsilon$ with $0 < \beta < D_1$ and $\delta_\varepsilon$ being a mean level spacing in the corresponding part of the spectrum. Such a choice allows us to get meaningful physical results for any system in a delocalized phase. Unless otherwise noted, we consider bulk with $\delta_\varepsilon \propto N^{-1}$.

The idea of the diagonal approximation is to say that, in a thermodynamic limit, the sum in the denominator of (18) is dominated by the diagonal elements of the reduced Green's function [45, 47]:

$$G_{ii}(\varepsilon) \xrightarrow[N \to \infty]{} \left( z - \varepsilon_i - \sum_{j \neq i} |V_{ij}|^2 G_{jj}^{(i)}(z) \right)^{-1}. \qquad (20)$$

The approximation is considered valid, provided the system is in a non-ergodic phase.[6] However, the SFD's graphical algebra introduced above can not be directly applied to this expression as it also contains complex variables. To proceed, we need to either generalize our graphical

---

[6]See Appendix B for more details on the diagonal cavity method applicability conditions.

algebra to complex random variables or write the local density of states explicitly staying in the real domain. While the former approach is also possible (see Appendix D), for simplicity, here we employ the latter one:

$$\nu_i(\varepsilon) \sim \frac{\Gamma_i(\varepsilon)}{(\varepsilon - \varepsilon_i)^2 + \Gamma_i(\varepsilon)^2}, \qquad \Gamma_i(\varepsilon) = \sum_{j \neq i} |V_{ij}|^2 \nu_j(\varepsilon). \tag{21}$$

In this expression, we neglected the real part of the diagonal self-energy $\Sigma_i = \sum_{j \neq i} |V_{ij}|^2 G_{jj}^{(i)}(\varepsilon)$ compared to the on-site disorder amplitude[7] and omitted $\eta$ compared to the broadening $\Gamma_i(\varepsilon)$. These simplifications, again, restrict the applicability of the following results to the non-ergodic delocalized part of the phase diagram. Now, the problem of calculating the spectrum of fractal dimensions of the local density of states $\nu_i(\varepsilon)$ appears as a self-consistent problem [31, 44, 45, 47], and below, we show how to solve it using the SFD graphical algebra introduced earlier.

First, let us consider the broadening $\Gamma_i(\varepsilon)$. Since the broadening can be expressed as an extensive sum, its SFD possesses all the characteristic properties of extensive sums, allowing us to guess the SFD almost completely. Indeed, since this is an extensive sum, there can be nothing to the right of $\alpha = \alpha_0(\Gamma_i)$ corresponding to the broadening's typical value, i.e., $f_{\Gamma_i}(\alpha > \alpha_0(\Gamma_i)) = -\infty$. On the other hand, the terms entering the sum do not correspond to any fat-tailed distribution: $V_{ij}$ is Gaussian by definition, and the LDOS is bounded from above by inverse amplitude of the on-site disorder, which is of the order of $N^0$. This means that there can be nothing also to the left of $\alpha_0(\Gamma_i)$, i.e., $f_{\Gamma_i}(\alpha < \alpha_0(\Gamma_i)) = -\infty$. As a result, its SFD should look like

$$\Gamma_i : \qquad \qquad \tag{22}$$

The only unknown parameter left is the value of this typical value $\Gamma_i \sim N^{-c}$, which we parameterized by $\alpha_0(\Gamma_i) = c$.

The site index $i$ is used in the previous paragraph in $\Gamma_i$ and $\nu_i$ to specify the random quantities, corresponding to this site. Due to the statistical homogeneity, assumed by the self-consistent cavity formulation of the problem and respected by the model under consideration, further, we omit this unnecessary (double) indexing and write just $\Gamma$, or $f_\Gamma(\alpha)$, or, similarly, $p_\nu(x)$, where possible. Therefore, the index-free symbols $\Gamma$, $\nu$, etc., should be considered as shortcuts for "level broadening", "local density of states", etc.

Now, having the shape of the broadening fixed, we can calculate the distribution of $\nu$, substitute it to the definition of $\Gamma$, and find $c$ self-consistently. To perform the first step, one should ensure that all terms entering the expression for $\nu$ are independent – which is not true because $\Gamma$ enters the expression two times. However, since, from the SFDs' point of view, $\Gamma$ is just a constant, we can still perform this step without introducing any further complications.

---

[7] The real part of the self-energy appears in the expression for LDOS as an energy shift renormalizing the on-site energies. Hence, the shift is not important until the renormalization affects the model's bandwidth in the ergodic phase, i.e. until $\gamma \leq 1$. This can be seen just by comparing the bandwidth $N^{(1-\gamma)/2}$ of the hopping matrix $V$ to the on-site disorder amplitude, which is 1.



The calculation is depicted below:

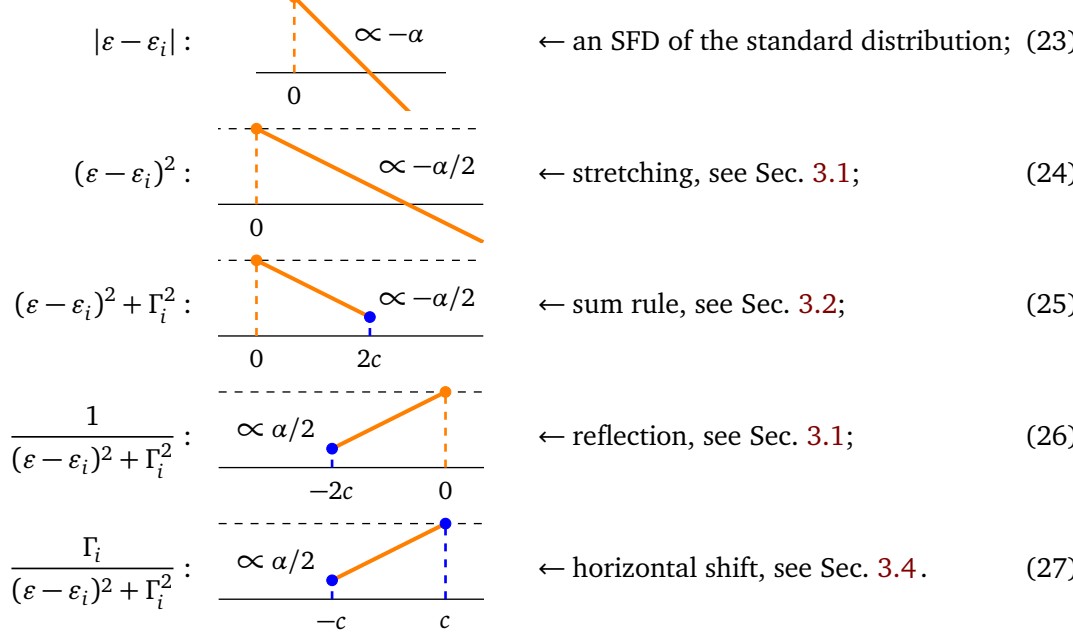

$|\varepsilon - \varepsilon_i|$ :    $\propto -\alpha$    ← an SFD of the standard distribution;    (23)

$(\varepsilon - \varepsilon_i)^2$ :    $\propto -\alpha/2$    ← stretching, see Sec. 3.1;    (24)

$(\varepsilon - \varepsilon_i)^2 + \Gamma_i^2$ :    $\propto -\alpha/2$    ← sum rule, see Sec. 3.2;    (25)

$\dfrac{1}{(\varepsilon - \varepsilon_i)^2 + \Gamma_i^2}$ :    $\propto \alpha/2$    ← reflection, see Sec. 3.1;    (26)

$\dfrac{\Gamma_i}{(\varepsilon - \varepsilon_i)^2 + \Gamma_i^2}$ :    $\propto \alpha/2$    ← horizontal shift, see Sec. 3.4.    (27)

As one can see from a comparison with the result obtained in [38], the shape of the SFD we have just found is correct. Now, let us write the self-consistency equation:

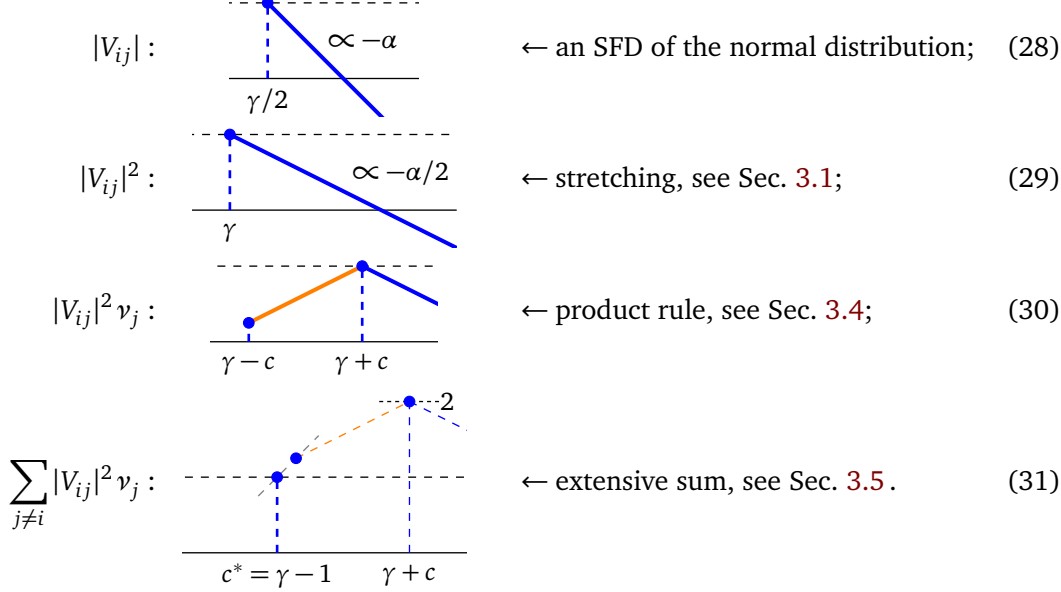

$|V_{ij}|$ :    $\propto -\alpha$    ← an SFD of the normal distribution;    (28)

$|V_{ij}|^2$ :    $\propto -\alpha/2$    ← stretching, see Sec. 3.1;    (29)

$|V_{ij}|^2 \nu_j$ :    ← product rule, see Sec. 3.4;    (30)

$\displaystyle\sum_{j \neq i} |V_{ij}|^2 \nu_j$ :    ← extensive sum, see Sec. 3.5.    (31)

In the last equation, (31), we demonstrated the elevation of the highest blue point with the small dashed line. Further we will use the same notation. Comparing (31) to (22), one can find $c = \gamma - 1$, in complete agreement with the previously known results. The ergodic and Anderson transitions then follow from the equations $c = 0$ ($\Gamma \propto N^0$, thus, all $\nu_i$ are roughly of the same amplitude) and $c = 1$ ($\Gamma \propto \delta_\varepsilon$ and, thus, the normalization of $\nu$ is contributed by a finite fraction of large components), giving, as expected,

$$\gamma_{ET} = 1, \quad \text{and} \quad \gamma_{AT} = 2. \qquad (32)$$

While the result (27) is not new, its graphical calculation already provides some insights. For example, the orange linear region with the slope $1/2$ between $\alpha = \pm c$ comes from the fact

that our diagonal matrix elements are homogeneous in the bulk of the distribution and, thus, this should be a general feature of many similar random matrix models.[8] The only contribution from the off-diagonal elements to the final result (27) was in the form of fixing the value of $c$, which is reflected by the different colors we use for different contributions. The shape itself was controlled only by the statistics of the on-site disorder and, in particular, by the fact that $p_{\varepsilon_i}(x)$ is finite and non-zero as $x \to \varepsilon$.

While due to the introduced above restriction $\Gamma \gg \eta \gg \delta_\varepsilon$, we cannot apply our method in the localized phase, $\gamma > 2$, let us see how the Anderson localization *formally* looks like from our graphical self-consistency equation's point of view. When $\gamma$ approaches 2 from below, the blue points at $\alpha = 2\gamma - 2 - c$ and $\alpha = \gamma - c$ from (31) approach each other until they coincide for $\gamma = \gamma_{AT} = 2$. For $\gamma > 2$, the same picture looks like

$$\sum_{j \neq i} |V_{ij}|^2 \nu_j : \qquad\qquad \gamma - c \quad \gamma + c - 2 \qquad \gamma + c \tag{33}$$

which already contradicts our initial guess (22) for the level broadening. Nevertheless, one can try to use (33) as a new guess, which results in a self-consistency equation $c = \gamma + c - 2$. The unknown $c$ drops out of this equation, leaving us with the only point this construction can hold for, $\gamma = 2$. And, while this attempt does not lead us to a correct solution, it hints at an important conclusion about the localized phase: the level broadening for $\gamma > 2$ no longer has the form (22), originated from the collective contribution of many sum's terms $|V_{ij}|^2 \nu_j$. Instead, as one may assume from the previously known results [38], it should also contain an individual contribution from the localized wave function's maximum to preserve the LDOS normalization condition.

# 5 Lévy Rosenzweig-Porter model

Inspired by the success of the Rosenzweig-Porter model with normally distributed off-diagonal amplitudes and by the distribution of the effective matrix elements in the Hilbert space, derived for many-body disordered models [49], a similar RP ensemble with the fat-tailed Lévy-distributed amplitudes was proposed [30, 31]. The main motivation was that this Lévy Rosenzweig-Porter should host the desired multifractal phase. This section reviews the statement and provides our own analysis of the proposed model.

The Lévy Rosenzweig-Porter ensemble is the ensemble of random matrices (17), with the uncorrelated diagonal on-site energies $\varepsilon_i$, distributed according to some narrow size-independent distribution like the normal or box distribution, and with the i.i.d. off-diagonal elements $V_{ij}$ of typical amplitudes $N^{-\gamma/2}$, distributed according to the following PDF with the parameter $\mu$:

$$p(V_{ij}) \sim \frac{2\mu N^{-\mu\gamma/2}}{|V_{ij}|^{1+\mu}} \theta\left(|V_{ij}| - N^{-\gamma/2}\right). \tag{34}$$

Such a polynomial decay of the PDF tails makes the off-diagonal elements' distribution heavy-tailed for $\mu < 2$ (variance is undefined) and fat-tailed for $\mu < 1$ (mean of the absolute value

---

[8]Recently, an interest to non-homogeneous (Cantor-set-like) on-site energies distributions has arisen [44, 48].

is undefined). The SFDs of the distributions with such polynomial tails look like

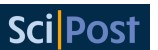

$$\tag{35}$$

These heavy- and fat-tail properties are precisely what led the authors of [31] to an elegant argument supporting the existence of the multifractal phase in the Lévy-RP models. The argument is based on estimating the fractal dimensions $D_1$ and $D_\infty$ and concludes that they are not equal, meaning the wave functions are multifractal.

Below, we calculate a fractal spectrum of the bulk eigenstates of the Lévy Rosenzweig-Porter (Lévy-RP) model, analogously to how we did it for the Gaussian Rosenzweig-Porter model in Sec. 4. As we are mostly interested in the (multi)fractal phase, which is expected [31] to exist for $1 < \mu < 2$ and $2/\mu < \gamma < 2$, this is exactly the parameter range we consider. Thus, we start by defining the fractal spectrum of $|V_{ij}|^2$ and trying to guess the SFD of the broadening $\Gamma$. Rescaling (35) according to the exponentiation rule from Sec. 3.1 and using $\alpha_0 = \gamma/2$, we get

$$|V_{ij}|^2 : \qquad\qquad \tag{36}$$

A product of $|V_{ij}|^2$ and the LDOS $\nu_j$ is at least as heavy-tailed as $|V_{ij}|^2$. Taking also into account that $\nu$ is normalizable, and, i.e., bounded, we conclude that the extensive sum $\Gamma_i = \sum_{j \neq i} |V_{ij}|^2 \nu_j$ must have the SFD of the form

$$\Gamma : \qquad\qquad \tag{37}$$

where the right-wing tails disappear, as usual, due to the effect of zero suppression in extensive sums. The only unknown here, like in the Gaussian RP case from Sec. 4, is the scaling exponent $c$ of the typical value of $\Gamma$.

Next, to obtain the SFD of $\nu$, we use the mix rule from Sec. 3.3. Indeed, taking for each fixed $\Gamma$-value the corresponding conditional distribution (27) and the distribution of 'weights', given by (37), we get

$$\nu : \qquad\qquad \tag{38}$$

where the new 'zeros' part to the right of $c$ originates from the rare realizations of $\Gamma_i$ so large that $\text{Im}[G_{ii}] \propto \Gamma_i^{-1}$. The label with the left arrow demonstrates that $f_\nu(c+0) = 1 - \mu c$. To finish the calculations, we write the self-consistency equation:

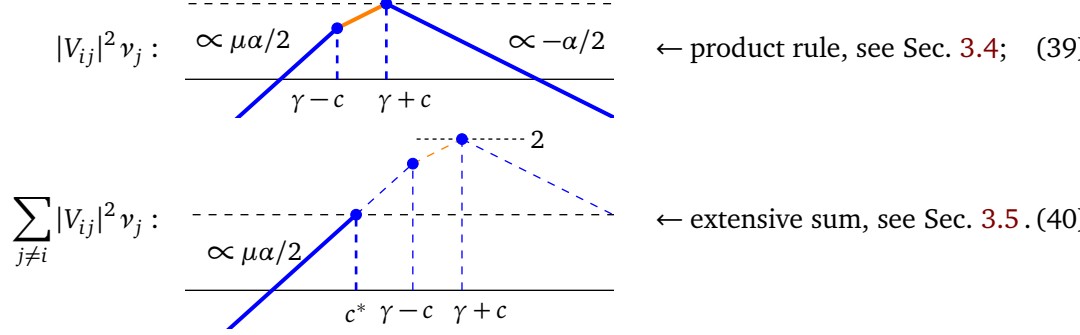

$$|V_{ij}|^2 \nu_j : \qquad\qquad \leftarrow \text{product rule, see Sec. 3.4;} \quad \tag{39}$$

$$\sum_{j \neq i} |V_{ij}|^2 \nu_j : \qquad\qquad \leftarrow \text{extensive sum, see Sec. 3.5.} \tag{40}$$

Here, $c^* = \gamma - c - (1-c) \cdot 2/\mu = c$, meaning that

$$\gamma_{eff} = c + 1 = \frac{(2+\gamma)\mu - 4}{2(\mu - 1)}. \tag{41}$$

As one can see from (38), the SFD of the LDOS in the Lévy-RP model, similarly to the Gaussian RP case, corresponds to such $f(\alpha)$ that $f(\alpha < \alpha_1) = -\infty$ and, hence, it is just fractal, not multifractal. Moreover, calculating the fractal dimensions $D_1$ and $D_\infty$ using this SFD[9] and the self-consistent value of $\gamma_{eff}$, we get

$$D_{1,\infty} = 1 - c = 2 - \gamma_{eff} = \frac{\mu(2-\gamma)}{2(\mu - 1)}, \quad 2/\mu < \gamma < 2, \quad 1 < \mu < 2, \tag{42}$$

which coincides with the results for $D_1$ from [31], but not with the ones for $D_\infty$. The fractal phase with $0 < D_1 < 1$, as in the Gaussian RP case from Sec. 4, gives a way to the ergodic one as soon as $c = 0$, provided

$$\gamma_{ET} = 2/\mu, \qquad 1 < \mu < 2, \tag{43}$$

while the localized phase appears when $c = 1$, giving

$$\gamma_{AT} = 2, \qquad 1 < \mu < 2. \tag{44}$$

As one can see from these equations, the purpose of introducing $\gamma_{eff}$ was in preserving the analogy with the Gaussian RP model, as the fractal dimension takes a universal form $D_{q > 1/2} = 2 - \gamma_{eff}$ as well as the corresponding phase diagram (32).

At this point, we are ready to draw a phase diagram of the Lévy-RP model according to its LDOS SFD. To do that, in addition to what we already did, we need to explore the regions $\mu > 2$ and $\mu < 1$. The former case of $\mu > 2$ reproduces the results of the Gaussian RP: It follows from the fact that the hopping distribution ceases to be heavy-tailed, the slope $\mu/2$ from (39) becomes larger than one, and, as a consequence, the extensive sum from (40) produces the same expression for $c^*$ as in the Gaussian RP model, giving the ergodic and the Anderson localization transitions at $\gamma_{ET} = 1$ and $\gamma_{AT} = 2$ for any $\mu > 2$. The latter case of $\mu < 1$ is a bit more interesting: similarly to the situation with $\gamma > 2$ from the end of Sec. 4, $c$ drops out of the corresponding self-consistency equation, giving the Anderson localization transition line as $\gamma_{AT} = 2/\mu$. However, the support set dimensions (42) on this line are now not zero but one, meaning that the line corresponds also to the ergodic transition. The resulting phase diagram is given in Fig. 7.

We already said that, due to the heavy-tailed distribution of the off-diagonal elements, the convergence of numerical simulations to the thermodynamic limit of the Lévy-RP model is very slow, see also [44]. However, an attempt to verify our analytical prediction can be seen in Fig. 8: an extrapolation to infinite sizes [12, 38, 50] is shown by black dots, and our theoretical prediction is by the thick red line(s). We have used different kinds and directions of extrapolation to diminish the finite-size effects in the most efficient manner, please see the caption of Fig. 8 for more details.

The meaning of the two red lines, dashed and solid, to the right of the typical value, $\alpha > \alpha_0 = 0.55$, is the following: the dashed line is plotted according to (38), while the solid line shows the actual behavior of the SFD in this region supported by the extrapolation of our numerical results. One can see that the analytical calculations, described above in Eq. (38), are confirmed by the numerical simulations for all $\alpha < \alpha_0$ below the typical value. The main difference between numerics and analytical predictions appears for $\alpha > \alpha_0$. Here one should mention that for the Lévy-RP model this issue does not affect the values of $D_q$ even for negative $q$ as in both cases of the behavior $f(\alpha > \alpha_0)$ the fractal dimension diverges at $q < -\mu/2$.

---

[9]To obtain the correct result, one should first normalize the local density of states to one (like if it would be a wave function) and only then use (2), (3), or Fig. 1.

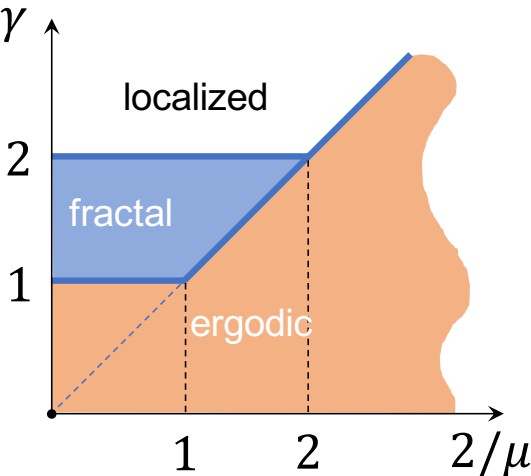

Figure 7: Localization phase diagram for the Lévy Rosenzweig-Porter model, according to its LDOS distribution.

Nevertheless, let us consider the reason why the calculations above failed to capture the behavior at $\alpha > \alpha_0$. This should originate from one of the approximations we made on the way. Here, we consider them one by one and point out which is the most crucial. Based on the estimation from Appendix B, we can conclude that the diagonal cavity approximation holds in the parameters range $2/\mu < \gamma < 2$, which we defined at the beginning of Sec. 5. This range corresponds to the non-ergodic extended phase and, thus, it is not this approximation, which is responsible for the inconsistency between (38) and simulation in Fig. 8.

As a result, the only other approximation which might be crucial is the one we made following [31]: it was throwing away the real part of the self-energy $\Sigma_i(\varepsilon - i\eta) = \sum_{j \neq i} |V_{ij}|^2 G_{jj}^{(i)}(\varepsilon - i\eta)$ compared to the diagonal disorder. This approximation seems reasonable until the typical value of $\text{Re}\,\Sigma_i$ is less than that of the on-site disorder and the self-energy distribution is less heavy-tailed than the on-site disorder. Indeed, provided the conditions above hold, the SFD of $|\varepsilon_i + \text{Re}\,\Sigma_i|$ will be identical to that of $|\varepsilon_i|$. This is not the case for the Lévy-RP model.

Note that, strictly speaking, to prove the above statement, one should generalize the notion of SFD from the distributions on the positive axis to the ones on the entire real axis (see Appendix C for details of this approach). However, here we provide the simpler reasoning. Indeed, first, the shape of SFD for the variable $|\varepsilon_i + \text{Re}\,\Sigma_i|$ at the atypically large values of $\text{Re}\,\Sigma_i \sim N^{-\alpha}$, $\alpha < \alpha_0(|\text{Re}\,\Sigma_i|)$, trivially coincides with that of $|\varepsilon_i| + |\text{Re}\,\Sigma_i|$, as one of the terms will dominate the sum and determine its sign. And second, the linear decay of the SFD with the unit slope at $\alpha > \alpha_0(|\text{Re}\,\Sigma_i|)$ follows from the finite value of the PDF for $|\varepsilon_i + \text{Re}\,\Sigma_i|$ at zero (analogously to the Porter-Thomas distribution in GOE/GUE ensembles), guaranteed, in turn, by the mutual independence of $\varepsilon_i$ and $\Sigma_i$. As a result, since the PDF of $|\varepsilon_i|$ is also finite at zero, one can neglect the difference between the SFDs of $|\varepsilon_i + \text{Re}\,\Sigma_i|$ and $|\varepsilon_i|$ provided $\alpha_0(|\varepsilon_i|) \leq \alpha_0(|\text{Re}\,\Sigma_i|)$ and the distribution of $|\text{Re}\,\Sigma_i|$ is less heavy-tailed than $|\varepsilon_i|$, i.e., $|\text{Re}\,\Sigma_i|$ 'never' dominates $|\varepsilon_i|$.

Returning back to the Lévy-RP after this comment, we have to conclude that the heavy-tailed hopping distribution violates the second condition because it *is* more heavy-tailed than the on-site disorder distribution. This prevents us from neglecting the real part of self-energy and is the source of the discrepancy at $\alpha > \alpha_0$.

More concretely, the above observation forces us to consider both real and imaginary parts of the self-energy or, equivalently, of Green's function $G_{ii}$, making it necessary to generalize our SFD algebra of independent r.v.s to that of two correlated ones or, equivalently, to the complex-

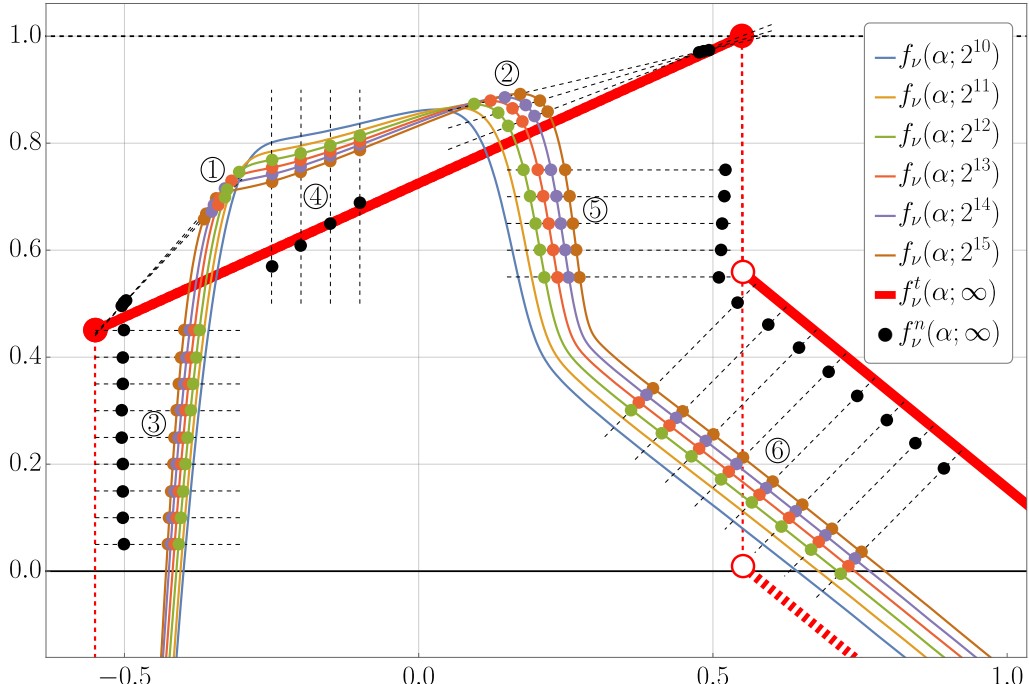

Figure 8: Spectrum of fractal dimensions of the LDOS of the Lévy-RP model for $\gamma = 1.6$, $\mu = 1.8$, $\gamma_{eff} = 1.55$, and several system sizes, including the thermodynamic limit theoretical prediction $f_\nu^t$ and extrapolation of numerical results $f_\nu^n$. A thick dashed red line shows the wrong prediction $f_\nu(c+0) = 1 - \mu c$ caused by us neglecting the real part of the self-energy. The correct prediction, $f_\nu(c+0) = 2 - \mu\gamma/2$, is shown by the solid red line and derived in Appendix E. For the extrapolation, we used two different techniques for ①–② and ③–⑥: in the former case, the straight lines are plotted through the points with the same derivative $q = f_\nu'(\alpha)$ for $q = -2, -1, 0, 2.5, 3$, and the lines' intersections are highlighted by the black points, while, in the latter case, the ansatz $f_\nu(\alpha(N); N) \sim f_\nu(\alpha; \infty) + A/\ln(N)$, $\alpha(N) \sim \alpha + B/\ln(N)$ is applied along the specified directions: at fixed $f(\alpha)$, $A = 0$, for ③, ⑤, at fixed $\alpha$, $B = 0$ for ④, and at a certain ratio $A/B$ for ⑥.

valued random variables. This broad and difficult topic is covered extensively in Appendix D, but the take-home message from that generalization is encouraging: the real part of the self-energy can only contribute to the atypically small values of LDOS, $\alpha > \alpha_0$. This fact can also be understood intuitively: while we increase the denominator of (21) keeping $\Gamma_i$ fixed, we decrease the value of the LDOS. The smallest value of the LDOS we get in such a way is its typical value, which is realized by a typical value of $|\varepsilon - \varepsilon_i| \sim O(1)$. Finally, as $\text{Re}\,\Sigma_i$ starts to dominate when it is larger than $O(1)$, it only contributes to the SFD part to the right of $\alpha_0$. Hence, we can continue neglecting it, provided we are only interested in $D_q$ with $q > 0$, given by $\alpha \leq \alpha_0$.

In order to compare our results, Eq. (38) and Fig. 8, with the previous claims, let us remind the results from [31]. In [31], the authors have considered in detail the miniband structure of the Lévy-RP model, and, both with the phenomenological arguments and the cavity method, derived the fractal dimension $D_1$, the results for which have been consistent from both methods as well as been numerically confirmed. The derivation of $D_\infty$ has been done only within the phenomenological arguments with the assumption that it is governed by a typical miniband broadening. It is well-known that the numerical studies of $D_q$ for large $q$ are usually very difficult due to the exponentially smaller statistics and very strong finite-size effects. Here,

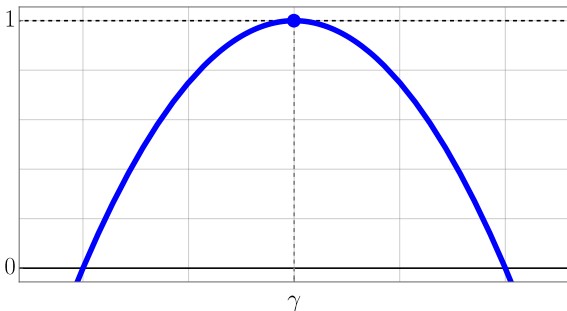

Figure 9: A spectrum of fractal dimensions of intensities of hopping matrix elements in the log-normal RP model.

we avoid having any phenomenological arguments and calculate the spectrum of fractal dimensions directly from the cavity method. The results show indeed the controversy to the phenomenological arguments of [31] for $q > 1$, including the typicality argument of the mini-band broadening for $q \to \infty$.

## 6 Log-normal Rosenzweig-Porter model

Another candidate to host a genuinely multifractal phase, circulating in the literature, is the so-called log-normal Rosenzweig-Porter model [3,18,22]. Its definition is analogous to other RP models, but the distribution of hopping matrix elements is defined as a real-valued log-normal distribution, with the parameters scaling with the system size $N$:

$$p_{V_{i \neq j}}(v) \propto \frac{1}{|v|} \exp\left\{-\frac{\ln^2(|v|/v_{typ})}{2p \ln\left(v_{typ}^{-1}\right)}\right\}, \quad v_{typ} = N^{-\gamma/2}. \tag{45}$$

Proceeding according to (1), we find that its SFD is parabolic, $f_{|V_{i \neq j}|^2}(\alpha) = 1 - (\alpha - \gamma)^2/(4p\gamma)$, and hence truly multifractal, see Fig. 9. Indeed, if we want a truly multifractal wave function, why not to start from a truly multifractal hopping distribution?

As we usually do at the beginning of the graphical calculations, let us define the range of parameters we consider and guess the first step's SFD. Defining the hopping SFD by the relation $f_{|V_{i \neq j}|^2}(\alpha) = 1 - (\alpha - \gamma)^2/(4a)$, let us start from considering $1 < \gamma < 2$ and small $a = p\gamma > 0$. The 'smallness' of $a$ will be defined later. But, considering that, for $a \to 0$, the hopping distribution approaches a narrow distribution with all moments well-defined, it is reasonable to assume that the LN-RP LDOS in this regime will be close to the Gaussian RP LDOS. With this idea in mind, let us start from the Gaussian RP LDOS SFD (27) and multiply it by the squared hopping element, with the log-normal distribution:

$$|V_{ij}|^2 \nu_j: \tag{46}$$

One can see how the orange segment with the slope $1/2$ appears smoothly in the blue parabola between the points $A$ and $B$, where the parabola's slope is also equal to $1/2$. Thus, the values of $A$, $B$, and $C = \alpha_0(|V_{ij}|^2 \nu_j)$ are, correspondingly, $\gamma - c - a$, $\gamma + c - a$, and $\gamma + c$. Next, we

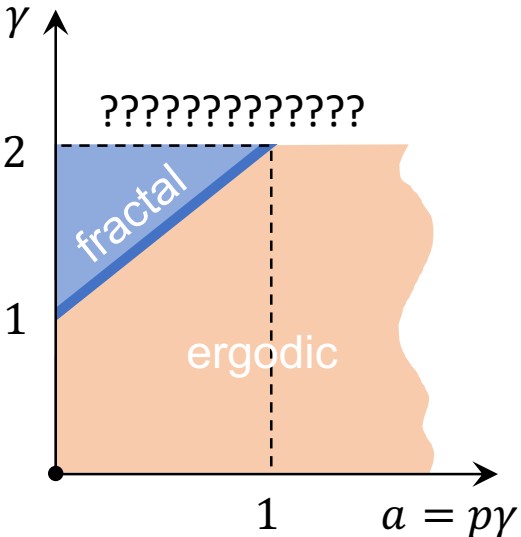

Figure 10: A part of the phase diagram for the LN-RP LDOS below $\gamma = 2$. The question marks signify the parameter range which we have not yet described.

calculate the level broadening

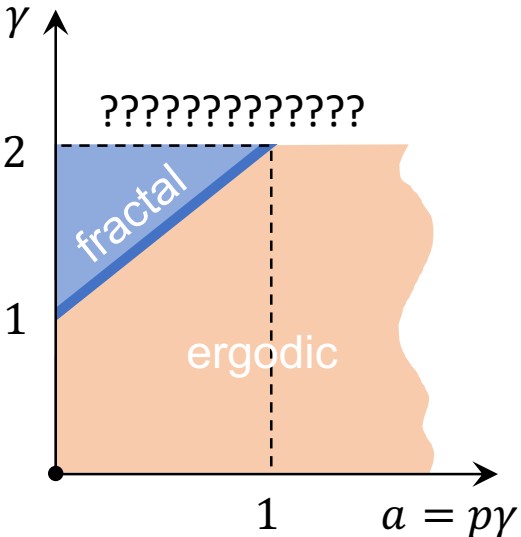

$$\sum_{j \neq i} |V_{ij}|^2 \nu_j : \tag{47}$$

The gray dot located at $X = \gamma - c - 2a$ marks the point on the SFD (46) where its slope is equal to one. The tangent line with a unit slope, touching this point, crosses the level $f(\alpha) = 1$ at the point $c^* = X - (1 - c - a)$. Substituting $c^* \to c$, we get

$$c = \gamma - a - 1, \quad \text{and} \quad \gamma_{eff} = \gamma - a. \tag{48}$$

From the shape of the SFD for the level broadening, we see that the resulting self-consistent LDOS SFD has the RP-like shape for $\alpha < \gamma_{eff}$. At the same time, for $\alpha > \gamma_{eff}$ it should show a very low probability of zeros. This is the only place where the parabolic input reveals itself. Formally speaking, this SFD is already multifractal for $D_q$ with negative $q$, but from the physical application perspective and according to the definition we gave at the end of Sec. 2, we consider it as a trivial fractal phase.

From Eq. (48), one can find the part of the phase diagram at $\gamma < 2$, Fig. 10. Indeed, the case of $c = 0$ corresponds to the level broadening of the order of the entire bandwidth which should correspond to the ergodic transition. This gives

$$\gamma_{ET} = a + 1 \iff \gamma_{ET} = 1/(1-p), \qquad p < 1/2. \tag{49}$$

In the r.h.s. we have used the standard substitution $a = p\gamma$.

let us now consider the question of the parameter's $a$ smallness. The geometric construction pictured in (47) holds only until $X > c^*$. This implies the restriction $\gamma < 2$. Given that, we can now draw a part of our model's phase diagram, Fig. 10.

Next, let us try to move to higher values of $\gamma$. To do that, let us start with some $\gamma$ in the fractal phase, $1 + a < \gamma < 2$, and gradually increase it until we reach the unexplored territory.

When the gray dot from Eq. (47) crosses the horizontal dashed line $f(\alpha) = 1$, we can no longer determine $c^*$ as a crossing point between the unit-slope tangent line and the level $f(\alpha) = 1$. Instead, we have to use the following construction:

$$\sum_{j\neq i} |V_{ij}|^2 \nu_j : \qquad\qquad\qquad\qquad \tag{50}$$

From this we find that $c^* = \gamma - c - 2\sqrt{a(1-c)}$, and the self-consistent solution for $c$ is now

$$c = \frac{\gamma - a - \sqrt{a(4 + a - 2\gamma)}}{2}. \tag{51}$$

Since the part of the SFD for the level broadening to the left of $c$ does not have even a single point with a slope less than or equal to $1/2$, the resulting SFD for the LDOS in the LN-RP model again has the shape of the LDOS SFD for the Gaussian RP, except for the part to the right of the typical value $\alpha = c$. Note that after the substitution of $a = p\gamma$ Eq. (51) agrees well with (49) in [3] and (C3) in [31] for $\tau^* \equiv \alpha(\gamma, p) \equiv c$.

The square root in the self-consistent expression (51) for $c$ immediately tells us that the result is valid only until $\gamma \le 2 + a/2$. From the geometrical point of view, the line $\gamma = 2 + a/2$ corresponds to the situation when the orange segment of (46) touches the level $f(\alpha) = 0$. Notice a similarity with the case $\gamma > 2$, discussed at the end of Sec. 4 in the context of the Gaussian RP model: the change in geometry happening for $\gamma > 2 + a/2$ calls to modify the self-consistency equation once more, but, if one tries to do it, one would quickly realize that it is impossible as $c$ drops out of the equation, leaving us with just the expression for this borderline itself. Similarly to the discussion at the end of Sec. 4, this disappearance of $c$ from the self-consistency equation hints that one of the basic conditions cannot be satisfied, namely, the wave-function normalization condition, leading to the emergence of the localization peak $f_\nu(\alpha = -1) = 0$ and signaling the Anderson transition. Thus, the expression for the Anderson transition from the fractal phase is given by

$$\gamma_{AT} = 2 + \frac{a}{2} \quad \Longleftrightarrow \quad \gamma_{AT} = \frac{2}{1 - p/2}, \tag{52}$$

where the latter expression is straightforwardly derived from $\gamma = 2 + a/2$ after the substitution $a = p\gamma$. Especially intriguing is that, like in the Lévy-RP model at $\gamma = 2/\mu$ and $\mu < 1$, the fractal dimensions on the Anderson transition lines are finite, implying a discontinuity of these quantities. Given that, in the LN-RP case, the line is the borderline of the fractal phase and, hence, the borderline of our method's applicability region, we cannot rule out the possibility of having a fine-tuned multifractality right on this line.

To finish the LN-RP diagram exploration, let us find the line corresponding to the ergodicity transition at $\gamma > 2$. To do that, we solve the equation $c = 0$ and get

$$\gamma_{ET} = 2\sqrt{a} \quad \Longleftrightarrow \quad \gamma_{ET} = 4p, \tag{53}$$

the latter form, again, is the known expression obtained from the former one by the substitution $a = p\gamma$. As a result, summarizing Eqs. (49), (52), and (53), we obtain the full phase diagram for the LN-RP model, see Fig. 11, confirming the previous results from [3,18,22,31]. It has a tricritical point analogous to the Lévy-RP case. According to [18], the case of RRG corresponds to the bisector $a = \gamma$ ($p = 1$) in this phase diagram, which crosses the above tricritical

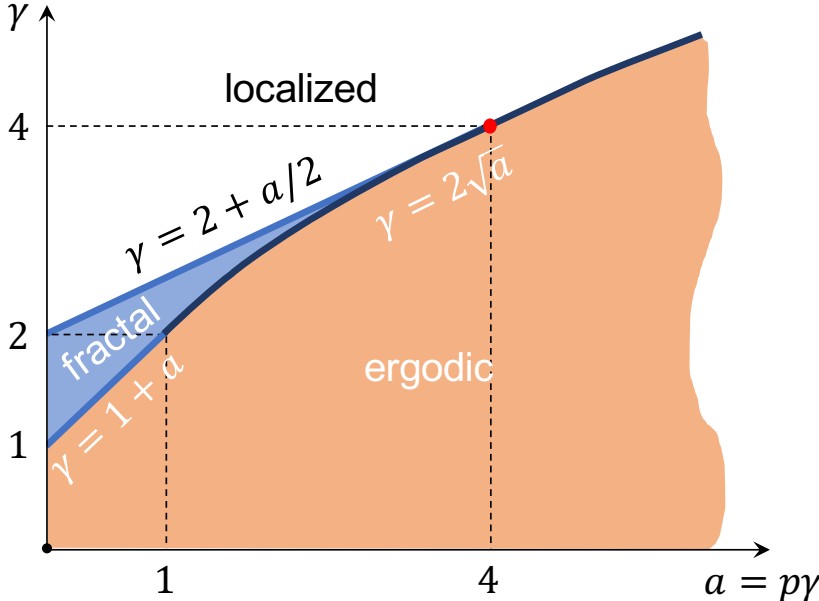

Figure 11: A full LN-RP LDOS phase diagram. The dark blue line shows where the expression $\gamma = 2\sqrt{a}$ is applicable, and the red point marks the tricritical point.

point $a = \gamma = 4$. And again, the diagram may only host multifractal states on the border of the localized phase. The rest of the diagram contains only localized, fractal, or ergodic phases.

But how is this possible that even the model with a multifractal distribution of its hopping elements failed to host a multifractal phase? One of the possible answers to this question lies in the observable we studied: recall that the local density of states, being an average over an extensive number of wave functions, does not necessarily reproduce all the details of the individual wave function distributions. Thus, as it was shown, e.g., for the Anderson model on the Cayley tree in [19,20], the absence of multifractality in the LDOS SFD does not eliminate the possibility of having the multifractal statistics of the eigenstates.

In order to examine this opportunity of having the wave-function multifractality together with the fractality of the LDOS, we consider the relation between their SFDs in the next section.

## 7 Relation between LDOS and eigenstate distributions

We are unaware of any method as powerful as the self-consistent cavity method but for individual wave functions $\psi_n(i)$ instead of the local density of states $\nu_i(\varepsilon)$. And, while we cannot calculate the corresponding SFD $f_{|\psi|^2}(\alpha)$ directly, we can infer restrictions for this function implied by the shape of $f_\nu(\alpha)$. Surprisingly, this analysis can be performed for any random Hamiltonian, and the result of this section goes beyond the Rosenzweig-Porter family.

By definition (19), the local density of states $\nu_i(\varepsilon)$ is proportional to the average of the squared wave functions' amplitudes $|\psi_n(i)|^2$ over the energy window $\eta$ around the energy $\varepsilon$. This energy window for the averaging is controlled by the Lorentzian function meaning that its tails decrease as $(\varepsilon - E_n)^{-2}$. First, for simplicity, let us consider a box kernel instead of the Lorentzian one and define the box LDOS $\tilde{\nu}_i(\varepsilon)$ as

$$\tilde{\nu}_i(\varepsilon) = \delta_\varepsilon^{-1} \left\langle |\psi_n(i)|^2 \right\rangle_{\varepsilon \pm \eta} = \delta_\varepsilon^{-1} \frac{\sum_{n=1}^N |\psi_n(i)|^2 \theta(\eta - |E_n - \varepsilon|)}{\sum_{n=1}^N \theta(\eta - |E_n - \varepsilon|)}. \tag{54}$$

Later, we return back to the standard Lorenzian LDOS.

From now on, let us fix a specific site $i$ of our system. In the RP family's models, all sites are statistically equivalent, but, in general, they do not have to be. Each realization of a random Hamiltonian $H$ will then produce a single realization of $\tilde{\nu}_i(\varepsilon) = \tilde{\nu}_i(\varepsilon; H)$ and of the order of $N^\beta = \eta/\delta_\varepsilon \gg 1$ realizations of $|\psi_n(i)|^2 = |\psi_n(i; H)|^2$ contributing to the value of $\tilde{\nu}_i(\varepsilon; H)$, Eq. (54). Our first task is to relate the distribution of $|\psi_n(i)^2|$ at $|\varepsilon - E_n| < \eta$ to the distribution of $\tilde{\nu}_i(\varepsilon)$. For this, we consider $0 < \beta \ll 1$ in order to assume that the wave functions in this small energy window are statistically equivalent.[10]

The values of $|\psi_n(i; H)|^2$ from each realization of $H$ may or may not be correlated. Hence, since our graphical language was developed only for independent random variables, we cannot directly apply the generalized central limit theorem from Sec. 3.5 to this case of Eq. (54). Having said that, let us consider the whole ensemble of $H$ and a set $\tilde{\Omega}_\alpha$ consisting of all $|\psi_n(i; H)|^2$ contributing to $\tilde{\nu}_i(\varepsilon; H)$ such that $\tilde{\nu}_i(\varepsilon; H) = N^{-\alpha}$:

$$\tilde{\Omega}_\alpha = \left\{ |\psi_n(i; H)|^2 \,\middle|\, |E_n - \varepsilon| < \eta \,,\; \tilde{\nu}_i(\varepsilon; H) = N^{-\alpha} \right\} . \tag{55}$$

The unconditional distribution $p_{|\psi|^2}(x)$ of $|\psi_n(i)|^2$ from our energy window can be obtained from the conditional distribution $p_{|\psi|^2}(x|x \in \tilde{\Omega}_\alpha)$ of $|\psi_n(i)|^2 \in \tilde{\Omega}_\alpha$ by a probability chain rule as

$$p_{|\psi|^2}(x) = \int p_{|\psi|^2}(x|x \in \tilde{\Omega}_\alpha) p_{\tilde{\nu}}(N^{-\alpha}) \mathrm{d}N^{-\alpha} . \tag{56}$$

Transposing from the probability density functions to the corresponding spectra of fractal dimensions, we recover the mix rule from Sec. 3.3:

$$f_{|\psi|^2}(\alpha) = \max_\xi \left\{ f_{|\psi|^2}(\alpha|N^{-\alpha} \in \tilde{\Omega}_\xi) + f_{\tilde{\nu}}(\xi) - 1 \right\} . \tag{57}$$

This formula directly relates the SFD of the eigenvalues $f_{|\psi|^2}(\alpha)$ to the SFD of the box LDOS $f_{\tilde{\nu}}(\xi)$.

Now, let us consider $f_{|\psi|^2}(\alpha|N^{-\alpha} \in \tilde{\Omega}_\xi)$ and infer what it may look like. First, from the definition of $\tilde{\Omega}_\xi$, Eq. (55), we know that $|\psi|^2$ from our conditional distribution $p_{|\psi|^2}(x|x \in \tilde{\Omega}_\xi)$ cannot exceed $N^{-\xi}\delta_\varepsilon = N^{-\xi-1}$, see Eq. (54). Indeed, otherwise, such $|\psi|^2$ would give rise to $\tilde{\nu} > N^{-\xi}$ at least for some realizations of $H$ related to $\tilde{\Omega}_\xi$. Second, we know that the point $\alpha = \xi - \ln_N \delta_\varepsilon = \xi + 1$, $f_{|\psi|^2}(\alpha) = 1$ belongs to $f_{|\psi|^2}(\alpha|N^{-\alpha} \in \tilde{\Omega}_\xi)$. Otherwise, we would get $\tilde{\nu} < N^{-\xi}$ for some $H$ contributing to our conditional distribution. Hence, the conditional SFD is constrained to have the following form:

$$|\psi_n(i)|^2 \in \tilde{\Omega}_\xi : \qquad \underset{1+\xi}{\rule{0pt}{1.5em}} \tag{58}$$

Here, the blue part of the graph is the necessary part: $f_{|\psi|^2}(\alpha|N^{-\alpha} \in \tilde{\Omega}_\xi)$ has to be equal to 1 at $\alpha = 1 + \xi$ and to $-\infty$ at $\alpha < 1 + \xi$. In the interval $\alpha > 1 + \xi$, our conditional SFD can have any possible shape allowed by our definition of SFD, $f_{|\psi|^2}(\alpha|N^{-\alpha} \in \tilde{\Omega}_\xi) \leq 1$. Several examples of that are depicted in (58) by the thin lines of different colors. Hence, applying the mix rule to this manifold of conditional SFDs, we get that the unconditioned SFD $f_{|\psi|^2}(\alpha)$ can differ from the corresponding SFD $f_{\tilde{\nu}}(\alpha)$ of the box LDOS, Eq. (54), only in the region $\alpha > \alpha_0(\tilde{\nu})$. To the left of this point, these two SFDs must coincide.[11]

---

[10]If they are not, the result still holds but has a different physical meaning. It then relates the distribution of LDOS to the marginal distribution of all eigenstate coefficients from the considered energy window.

[11]Here we have considered the convex $f_{\tilde{\nu}}(\alpha)$. For non-convex ones, the difference may appear to the right, $\alpha > \alpha_*$, of each maximum $f_{\tilde{\nu}}(\alpha_*) = f_*$, but with the deviated values $f_{|\psi|^2}(\alpha) \leq f_*$.

Finally, let us return back to the proper LDOS with the Lorentzian kernel. It differs from the box LDOS, which we have just examined, because it can be dominated, in principle, by the wave functions outside the energy window $\varepsilon \pm \eta$. This possibility lifts the restriction for the point $\{1 + \xi, 1\}$ to belong to $f_{|\psi|^2}(\alpha|N^{-\alpha} \in \tilde{\Omega}_\xi)$ for all $\xi$ except $\xi = \alpha_1(v) - 1 \equiv D_1(v) - 1$. Indeed, the latter is just a consequence of the wave-function normalization condition. Thus, we arrive at the following conclusion about the relation between the distributions of the eigen-functions and the local density of states:

$$f_{|\psi|^2}(D_1) = f_v(D_1 - 1), \quad \text{and} \quad f_{|\psi|^2}(\alpha) \leq f_v(\alpha - 1), \qquad \alpha < \alpha_0(v). \tag{59}$$

For the log-normal and Lévy Rosenzweig-Porter models, it means that $D_q(|\psi|^2) = D_q(v)$ for $q \geq 1/2$. This conclusion follows from the fact that, for both of these models, $f_v(\alpha < D_1(v)) = -\infty$, while the derivative of $f_v(\alpha)$ at $\alpha = D_1(v) + 0$ is equal to $1/2$. For most practical applications, where only $q > 1/2$ matters, these models show only fractal, but not multifractal properties.

## 8 Absence of multifractality in Rosenzweig-Porter models

As one may have already guessed from the previously considered models, the finding of a multifractal phase hosted by an RP model is far from trivial. In this section, we will prove that it is, in fact, impossible.

Before proceeding to the proof itself, let us focus on an important limitation of our method. Indeed, as one can guess, the above-developed method is insensitive to the changes in the PDF that do not affect the SFD. In this sense, all the sparse graph models and the standard Anderson models on the lattices cannot be described by this method, as the localization transition and the fractality are not governed by the scaling with the system size $N$.

As a remarkable example, let us consider a random Hamiltonian $H = H_{RP} + A_{ER}$, where $H_{RP}$ is a Gaussian RP model Hamiltonian from (17), and $A_{ER}$ is an adjacency matrix of a random Erdos-Rényi graph, with a fluctuating finite number of non-zero hopping terms of the order of unity. The hopping of this 'Erdos-Rényi-RP model' corresponds to the SFD

$$|V_{ij}|^2 : \qquad \propto -\alpha/2 \tag{60}$$

hereafter, let us assume $\gamma > 1 + c$ with a certain positive $c > 0$. Note that the additional blue point at the origin in the above plot corresponds to a finite number of "neighbors" for any site of this model, connected to it by the hopping of the order one. This is given in addition to the all-to-all RP-like hopping of the scaling with the system size amplitude $N^{-\gamma/2}$, see, e.g., [51] for the case of correlated hopping of this kind. Performing the calculations of the SFD for $v$, one can straightforwardly show that for $\gamma > 1 + c$ *any* SFD curve, respecting Mirlin-Fyodorov symmetry [52], $f_v(\alpha) = \alpha + f_v(-\alpha)$, which is finite only on the support $|\alpha| < c$, i.e., $f_v(|\alpha| > c) = -\infty$, satisfies the self-consistent cavity equation for such hopping. For example, one can take the parabolic one $f_v(\alpha) = 1 - (\alpha - c)^2/4c$ for $|\alpha| < c$:

$$v : \tag{61}$$

What's the catch? As we have mentioned above, for the Anderson model on sparse random graphs, the hopping scaling is not the only thing that matters for the localization and ergodicity breaking. In principle, different on-site energy distributions with the same SFD will lead to

different LDOS SFDs, see, e.g., [17] for the RRG. Analogously, the different lattice dimensionalities of the standard Anderson model drastically change the localization diagram [53,54]. Thus, from the perspective of Laplace's method in its leading order, the problem is ill-defined for such models, as it is not the SFD of the off-diagonal matrix elements and $N$-scaling, but PDFs and prefactors that resolve the localization phase diagram. This leads to the solution's ambiguity within the above graphical method and its inapplicability to such problems. In the following paragraphs, we focus on the models, where a multifractal segment in the LDOS SFD necessarily originates from the hopping SFD. This assumption implies that the resulting solution is solely determined by the SFDs of hopping terms and on-site energies.

That being said, let us prove that conventional RP-like models, i.e., the models differing from the Gaussian RP only by the distribution of the i.i.d. uncorrelated hopping elements without the Erdos-Rényi component, cannot host any multifractal phase. To do that, we are going to exploit a similar anatomical approach to what we used in Sec. 7: we assume the solution found, trace back its features to the input distributions, and conclude if such a self-consistent solution can actually exist or not.

So, for concreteness, let us start with the shape of the LDOS SFD given, e.g., by (61). As will be shortly seen, this particular choice does not affect the argument. Being a part of the iteration procedure, this shape ought to originate from mixing together different Gaussian-RP-like Lorenzian shapes of LDOS (27) corresponding to different fixed values of $\Gamma$: please see the orange dashed line in (62) for $\Gamma \sim N^{-\xi}$ and recall, e.g., how we obtained (38). Schematically, this inheritance can be illustrated by

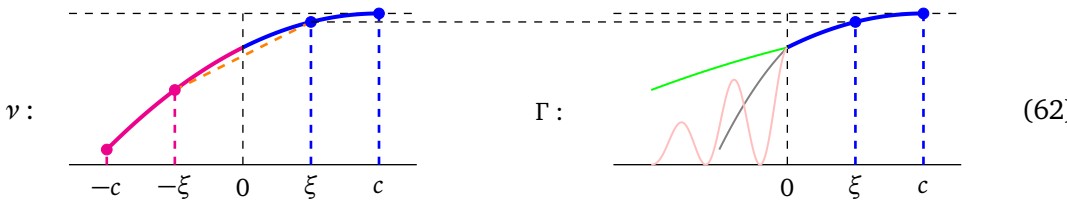

$$\tag{62}$$

here, the blue part of the LDOS SFD corresponds to the blue part of the broadening SFD, and, since, due to the Mirlin-Fyodorov symmetry, the magenta part of $f_\nu(\alpha)$ is controlled by the same blue part of $f_\Gamma(\alpha)$, the thin differently-colored curves of the SFD for the broadening demonstrate the unimportance of $f_\Gamma(\alpha < 0)$ as it can only affect $f_\nu(\alpha > c)$. The fact that the blue region of the broadening SFD produces the identical region on the LDOS SFD is due to the derivative of $f_\Gamma(\alpha)$ in this region being smaller than $1/2$. Indeed, as soon as it becomes larger than $1/2$, the corresponding contribution becomes subdominant with respect to the contribution of independent on-site energies, leading to the orange straight lines $\propto \alpha/2$ of the Poisson distribution we have already used to.

In its turn, the broadening SFD is obtained from the SFD of $|V_{ij}|^2 \nu_j$ by the extensive summation according to Sec. 3.5. Because of the zeros suppression effect, its tail, $f_\Gamma(\alpha < \alpha_0(\Gamma))$, may only originate from the tail of $|V_{ij}|^2 \nu_j$ such that, for $\alpha < c$, $f_\Gamma(\alpha) = f_{|V|^2\nu}(\alpha) + 1$. In our case, it must look like

$$|V_{ij}|^2 \nu_j : \qquad\qquad\qquad \tag{63}$$

here, as usual, the solid horizontal axis marks the level $f(\alpha) = 0$. What happens to the right of $\alpha = c$ is, again, unimportant, as long as $c$ stays the typical value of the extensive sum $\sum_{j \neq i} |V_{ij}|^2 \nu_j$. Then, what the hopping distribution of such a model should be?

To answer this question, we need to consider the product rule from Sec. 3.4. According to this rule, we know that, if the SFD of the product contains a point with some derivative, a corresponding point with the same derivative must be present on at least one multipliers' SFD. Moreover, if both multipliers contain points with the same derivative, the corresponding point on the product's SFD has a well-defined value and position, e.g., in our case,

$$\alpha_q(|V|^2 \nu) = \alpha_q(|V|^2) + \alpha_q(\nu), \quad f_{|V|^2 \nu}(\alpha_q(|V|^2 \nu)) = f_{|V|^2}(\alpha_q(|V|^2)) + f_\nu(\alpha_q(\nu)) - 1. \quad (64)$$

From (62) and (63), we know that, for any point from the blue region, $\alpha_q(|V|^2 \nu) = \alpha_q(\nu)$, and $f_{|V|^2 \nu}(\alpha_q(|V|^2 \nu)) = f_\nu(\alpha_q(\nu)) - 1$, leaving us with the requirement for $f_{|V|^2}(\alpha)$ to pass through the point $\{0, 0\}$ and to have a discontinuous first derivative at this point. But it means that the multifractal shape of $f_\nu(\alpha)$ we assumed in our example originated not from the hopping or on-site energies SFDs but from something more subtle like specific PDFs or prefactors, i.e., it takes us beyond the conventional RP models' family and, thus, proves the absence of LDOS multifractality inside it.[12] Finally, we use the results of Sec. 7 to conclude that eigenstates of RP-like models, following the LDOS, also cannot be multifractal from the point of view of fractal dimensions $D_q$ with $q \geq 1/2$.

## 9  Conclusions and discussion

To sum up, in this paper, we have developed a powerful graphical method to calculate the multifractal properties, such as the spectrum of fractal dimensions and self-energies of the local density of states, in various models of the Rosenzweig-Porter type. We consider the Gaussian, Lévy, and log-normal Rosenzweig-Porter models and, with our method, easily reproduce their phase diagrams and fractal dimensions $D_1$.

In addition to that, we have calculated the entire spectra of fractal dimensions $f(\alpha)$ for the local density of states and found its relation to the one for the eigenstate coefficients. As a result, we have explicitly shown that, in all such models, the phase diagram may suggest the only type of the non-ergodic extended phase, which is fractal, but not multifractal, if we are speaking about the positive integer orders $q > 0$ of the fractal dimension $D_q$. The only formally multifractal part in $f(\alpha)$ we have found is beyond the point with the tangent slope $1/2$, $\alpha > \alpha_{1/2}$. This corresponds to the small or even negative moments $q < 1/2$ of the fractal dimensions $D_q$.

Having these calculations, we have managed to track back the origin of all parts of the spectrum of fractal dimensions and concluded that the uncorrelated models with i.i.d. hopping terms and conventional Poisson disorder can host only fractal phases. This statement can be, in principle, generalized to the Erdös-Rényi graphs with random hopping amplitudes and finite fraction $p \sim O(1)$ of non-zero edges. Statistically non-homogeneous distributions of the on-site disorder (like in [44]) may lead to a zero fraction of multifractal states, but the question if it can lead to the formation of an entire multifractal phase remains open. Another possibility for creating genuine multifractality is adding hopping-term [55] and on-site disorder correlations [56,57]. In principle, such correlations can be taken into account by combining the ideas of the Sherman-Morrison formula [46] with the graphical method, developed in this work.

The absence of multifractality, though, does not exclude non-trivial and anomalously slow dynamics in Rosenzweig-Porter models, see, e.g., [3, 30], which has a direct application to many-body disordered systems close to the MBL transition. The generalization of the developed graphical method for the effects of correlations of the local density of states may become

---

[12]We, of course, do not consider the cases with the on-site energy distributions described by PDFs that are non-analytic on some finite interval. While such cases may lead to multifractal LDOS SFD, we do not know any physical model that such exotic on-site energies' distributions can describe.

a good way to map such many-body systems to their random-matrix proxies. In this direction, one of the most prominent things is to focus on the frozen dynamical phase, suggested in [3], where the return probability of the wave-packet spreading can be stuck after some finite-time evolution.

Another direction to look at with the developed method is to consider a generic multifractal Rosenzweig-Porter model, obeying the RRG symmetry (see Eq. (6) in [3]) and focus on the origin of the tricritical point, found in the phase diagrams of Lévy- and log-normal RP models.

Finally, since our approach relies on Laplace's method of approximate integration, it does not only lead to a self-consistent solution in the thermodynamic limit $N \to \infty$ (which has been done in this paper) but also may allow calculating sub-leading orders of the finite-size scaling for $N \gg 1$. In this case, the cavity equation is not supposed to be solved self-consistently but to be viewed as a generator of the RG flow going to the known self-consistent fixed point. Among others, this point of view provides a way to analyze how to minimize the finite-size effects and speed up the convergence of numerical methods.

# Acknowledgments

We are grateful to V. E. Kravtsov for fruitful discussions and to him together with B. L. Altshuler and L. B. Ioffe for the works on the related topics.

**Funding information** I. M. K. acknowledges the support by the Russian Science Foundation, Grant No. 21-12-00409.

# A Extensive sum of non-negative i.i.d. r.v.s

Consider a non-negative random variable $X$ defined by a fractal spectrum $f_X(\alpha)$ and a random variable $S$ defined as a sum of $N^\beta$ independent copies of the random variable $X$:

$$s = \sum_{i=1}^{N^\beta} x_i \,. \tag{A.1}$$

Our goal here is to calculate $f_S(\alpha)$.

We start by introducing a size-independent coarse-graining of the horizontal axis in $\alpha$. Thus, we will not make any difference between $x_p$ and $x_q$ if both $-\log_N(x_p)$ and $-\log_N(x_q)$ lie between $\alpha_i$ and $\alpha_{i+1} = \alpha_i + \Delta\alpha$. After that, we take a single realization of $S$ or, equivalently, $N^\beta$ realizations of $X$, and count how many $X$'s in this particular sample correspond to the specific $\alpha_i$:

$$n_i = \#(x_i | \alpha_i < -\log_N(x_i) < \alpha_i + \Delta\alpha) \,. \tag{A.2}$$

Finally, we approximate the value of $S$ using the above empirical counts $n_i$ and the corresponding empirical fractal spectrum $g_i$ defined via $n_i = N^{g_i} \Delta\alpha$:

$$s \propto \sum_i n_i N^{-\alpha_i} \propto \sum_i N^{g_i - \alpha_i} \Delta\alpha \propto N^{\max_i (g_i - \alpha_i)} \Delta\alpha \,. \tag{A.3}$$

The result is closely related to Laplace's method of approximate integration: the point corresponding to the maximum of $g_i - \alpha_i$ lies close to the point where the line with unit slope touches the histogram $g_i$, see Fig. 12. We will use this last expression in all the following constructions as it allows a very natural approach to the probability distribution of $S$. Namely,

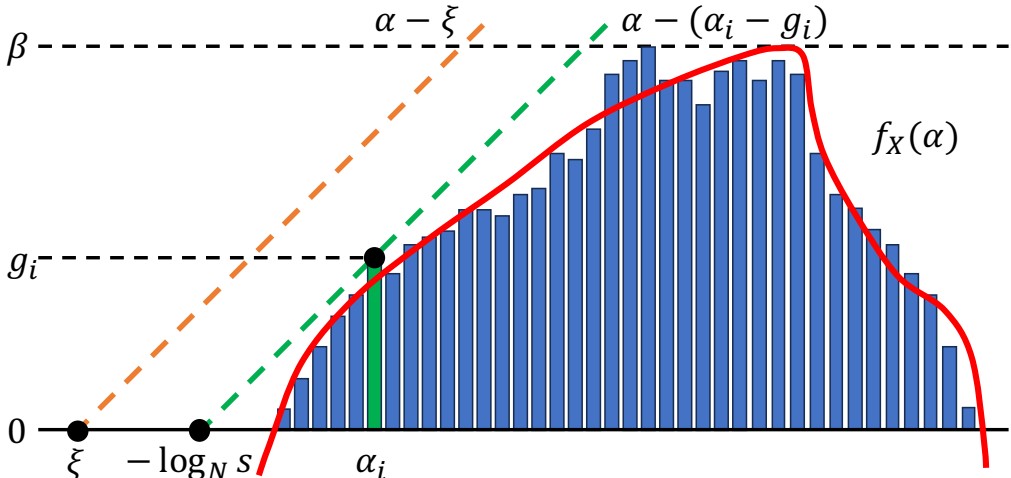

Figure 12: A given spectrum of fractal dimensions $f(\alpha)$ (in red) and one of the possible realizations of the coarse-grained empirical SFD histogram $g_i$. The dashed green line with the unit slope shows the value of $s$ resulting from this particular histogram realization. The fact that this particular realization of $s$ is smaller than $N^{-\xi}$ follows from that the blue histogram lies below the dashed orange line with unit slope $\alpha - \xi$. The plotted configuration illustrates the case when $\alpha - \xi > f_X(\alpha) - 1 + \beta$ for all $\alpha > \xi$. The opposite case with $\alpha - \xi < f_X(\alpha) - 1 + \beta$ for all $\alpha > \xi$ differs by the fact that, instead of only one bin being overfilled, it requires multiple bins being simultaneously underfilled, which is much less probable and leads to the (double-)exponential suppression of zeros, $f_S(\alpha > \alpha_0) = -\infty$.

we can define a probability that $s$ is less than $N^{-\xi}$ as a probability that all $g_i$ lie below the line $\alpha - \xi$, see Fig. 12. So now, we should calculate the likelihood of the empirical counts $n_i$ to be described by a vector $\boldsymbol{n}$, and then sum up the probabilities of all realizations respecting the condition above:

$$P(s < N^{-\xi}) = \sum_{0 \le n_i < N^{\alpha_i - \xi} \Delta \alpha} P\left(\boldsymbol{n} \,\Big|\, \sum_i n_i = N^{\beta}\right). \tag{A.4}$$

In principle, this approach could work, but in practice, it probably does not because of the total count conservation condition $\sum_i n_i = N^{\beta}$. Indeed, given that the probability to have $x \propto N^{-\alpha_i}$ is $p_i \propto N^{f_X(\alpha_i)-1}\Delta\alpha$, the probability $P(\boldsymbol{n}|\sum_i n_i = N^{\beta})$ itself is easy to write down:

$$P\left(\boldsymbol{n} \,\Big|\, \sum_i n_i = N^{\beta}\right) = N^{\beta}! \prod_i \frac{p_i^{n_i}}{n_i!}. \tag{A.5}$$

However, if we want to sum it over different $\boldsymbol{n}$, we would have a hard time doing that because of the factor $N^{\beta}!$ and inability to sum up all factors $p_i^{n_i}/n_i!$ independently. That is why, following the ideas of classical statistical mechanics, we now introduce a variable "number of particles" $\Sigma = \Sigma_i n_i$ and a narrow distribution over different values of $\Sigma$ peaked at $\Sigma = N^{\beta}$. Choosing this distribution to be the Poisson distribution

$$P(\Sigma; \beta) = \frac{(N^{\beta})^{\Sigma} e^{-N^{\beta}}}{\Sigma!}, \tag{A.6}$$

we cancel the nasty factorial and leave ourselves with the much easier expression to operate

with:

$$P(\boldsymbol{n};\beta) = P(\Sigma;\beta)P\left(\boldsymbol{n}\,\Big|\,\sum_i n_i = \Sigma\right) = \frac{(N^\beta)^\Sigma e^{-N^\beta}}{\Sigma!}\Sigma!\prod_i \frac{p_i^{n_i}}{n_i!} = e^{-N^\beta}\prod_i \frac{\nu_i^{n_i}}{n_i!}, \qquad (A.7)$$

here, we introduced $\nu_i = p_i N^\beta$ as the expected filling of the $i$'th bin. For sufficiently large $N^\beta$, the Poisson weight (A.6) becomes asymptotically close to a Gaussian one with mean $N^\beta$ and the standard deviation $N^{\beta/2}$, meaning that the probabilities of having $\boldsymbol{n}$ such that $\Sigma_i n_i = N^{\beta'}$ with $\beta' \neq \beta$ are double-exponentially suppressed in the thermodynamic limit. Hence, being interested only in the large-$N^\beta$ scaling behavior of the distribution of $S$, we can use (A.7) instead of (A.5) and write $P(s < N^{-\xi})$ from (A.4) in a large-$N^\beta$ limit as

$$\begin{aligned}
P(s < N^{-\xi}) &\propto \sum_{n_i < M_i(\xi)} P(\boldsymbol{n};\beta) = e^{-N^\beta}\prod_{i|M_i(\xi)>0}\sum_{n_i=0}^{M_i(\xi)}\frac{\nu_i^{n_i}}{n_i!} \\
&= e^{-N^\beta}\prod_{i|M_i(\xi)>0} e^{\nu_i}\frac{\Gamma(1+M_i(\xi),\nu_i)}{\Gamma(1+M_i(\xi))} \\
&= e^{-N^\beta\overline{p}(\xi)}\prod_{i|M_i(\xi)>0}\frac{\Gamma(1+M_i(\xi),\nu_i)}{\Gamma(1+M_i(\xi))},
\end{aligned} \qquad (A.8)$$

where $M_i(\xi) = N^{\alpha_i-\xi}\Delta\alpha$ is the largest possible count in the bin $i$, obeying the condition $s < N^{-\xi}$, $\Gamma(1+M_i(\xi),\nu_i)$ is the upper incomplete Gamma functions, and

$$\overline{p}(\xi) = 1 - \sum_{i|M_i(\xi)>0} p_i, \qquad (A.9)$$

is a probability of all "prohibited" for the given $\xi$ events. The only thing left to do now is to carefully write down the asymptotic expressions of the Gamma functions and go to the thermodynamic limit.

First, consider the case when $\alpha - \xi > f_X(\alpha) - 1 + \beta$ for all $\alpha > \xi$ or, in other words, when $M_i(\xi) \gg \nu_i$ for all $i|M_i(\xi) > 0$, see, again, Fig. 12. In this limit, we can write

$$\begin{aligned}
\prod_{i|M_i(\xi)>0}\frac{\Gamma(1+M_i(\xi),\nu_i)}{\Gamma(1+M_i(\xi))} &= \prod_{i|M_i(\xi)>0}\left(1 - \frac{\gamma(1+M_i(\xi),\nu_i)}{\Gamma(1+M_i(\xi))}\right) \\
&\sim \exp\left(-\sum_{i|M_i(\xi)>0}\frac{\gamma(1+M_i(\xi),\nu_i)}{\Gamma(1+M_i(\xi))}\right),
\end{aligned} \qquad (A.10)$$

where $\gamma(1+M_i(\xi),\nu_i)$ is the lower incomplete Gamma function. In the given limit, all terms in the sum are exponentially small:

$$\frac{\gamma(1+M_i(\xi),\nu_i)}{\Gamma(1+M_i(\xi))} \sim \frac{e^{M_i(\xi)-\nu_i}}{\sqrt{2\pi M_i(\xi)}}\left(\frac{\nu_i}{M_i(\xi)}\right)^{1+M_i(\xi)}. \qquad (A.11)$$

Moreover, since $\Delta\alpha$ does not scale with $N$ and the number of terms stays the same as $N \to \infty$, and because of this increasingly small $M_i(\xi)^{-M_i(\xi)}$ factor, each subsequent term is exponentially smaller than the previous one. In other words,

$$\sum_{i|M_i(\xi)>0}\frac{\gamma(1+M_i(\xi),\nu_i)}{\Gamma(1+M_i(\xi))} \sim \frac{\gamma(2,\nu(1))}{\Gamma(2)} \sim \frac{\nu(1)^2}{2}, \qquad (A.12)$$

where $v(1) = N^{f_X(\alpha_i)-1+\beta}\Delta\alpha$ is the expected filling at the point where $M_i(\xi) = 1$, or, in other words, at $\alpha_i = \xi - \log_N(\Delta\alpha) \to \xi$. Since in this case $f_X(\xi) - 1 + \beta < 0$, $v(1) \ll 1$, and

$$\prod_{i|M_i(\xi)>0} \frac{\Gamma(1 + M_i(\xi), v_i)}{\Gamma(1 + M_i(\xi))} \sim 1 - N^{2(f_X(\xi)-1+\beta)}/2. \tag{A.13}$$

Taking into account also the factor $e^{-N^\beta \overline{p}(\xi)}$, we get

$$\begin{aligned}
N^{f_S(\xi)-1} \propto \frac{\mathrm{d}}{\mathrm{d}x} P(s < N^{-\xi}) &\propto \frac{\mathrm{d}}{\mathrm{d}x} e^{-N^\beta \overline{p}(\xi)}(1 - N^{2(f(\xi)-1+\beta)}/2) \\
&\propto N^{f_X(\xi)-1+\beta} + N^{2(f(\xi)-1+\beta)} \propto N^{f_X(\xi)-1+\beta},
\end{aligned} \tag{A.14}$$

and, thus, when $\alpha - \xi > f_X(\alpha) - 1 + \beta$ for all $\alpha > \xi$,

$$f_S(\alpha) = f_X(\alpha) + \beta. \tag{A.15}$$

Now, consider the second case, i.e., when the line $\alpha - \xi$ has one or more intersections with $f_X(\alpha) - 1 + \beta$ in the region $\alpha > \xi$. Because in this case, there are such $i$ at $M_i(\xi) > 0$ that $v_i \gg M_i(\xi)$, the probability $P(s < N^{-\xi})$ becomes exponentially small. As written at the end of the caption to Fig. 12, this can be understood in simple entropic terms. Alternatively, it can be extracted from (A.8) and (A.9), using the Gamma function asymptotic, similarly to how we have done it for the intersection-free case. Moreover, as $\xi$ and $f_X(\xi)$ enter all the expressions as the exponents of $N$, this smallness is, in fact, double exponential, meaning that $P(s < N^{-\xi}) \propto \exp(N^{-\xi})$. Thus, without any involved calculations, we can say that, in this case, $f_S(\alpha) \to -\infty$ as $N \to \infty$. This result manifests the absence of zeros for extensive sums of non-negative random variables, which agrees with the suppression of zeros reported for finite sums in Sec 3.2.

# B  Applicability of the diagonal cavity approximation to the Lévy-RP model

The diagonal cavity approximation consists in substituting the exact self-energy $\sum_{j,k \neq i} V_{ij} G^{(i)}_{jk} V_{ki}$ with the diagonal cavity self-energy $\sum_{j \neq i} G^{(i)}_{jj} |V_{ij}|^2$. This approximation relies on the assumption that, for non-ergodic states, typical off-diagonal terms of the Green's function $G$ are negligible compared to the diagonal ones, see, e.g., [45, 47]. Such assumption is indeed quite reasonable, and it can be seen from the following simple picture: if all eigenstates are non-ergodic and, hence, each of them occupies $N^{D_1}$ sites, which is a measure zero of all sites for $D_1 < 1$, then the probability to find $\langle i|n\rangle \langle n|j\rangle$ of the order of $N^{-D_1}$ is of the order of $N^{-2(1-D_1)} \ll 1$. In other words, with the probability $N^0$ this off-diagonal projector element is much smaller than the local density of states on the support set. However, since there are many more off-diagonal elements than diagonal ones, let us proceed to more rigorous reasoning.

To obtain the (asymptotic) equation (20), all we need to do is to neglect, for some reason, the off-diagonal terms in the *exact* block matrix inversion formula (18). This simplification was used in numerous papers, including [31, 45, 58–61]. However, it is not so simple to justify the simplification in each particular case. Following [59] and [58], one may write the analogous to (18) expression for the off-diagonal Green's function elements

$$G_{ij} = -G_{ii} \sum_{k \neq i} V_{ik} G^{(i)}_{kj}, \tag{B.1}$$

and try to estimate the off-diagonal contribution based on the generalized central limit theorem. Assume the off-diagonal contribution is negligible: for $V_{ij}$ following a distribution with the PDF $p(x)$ decreasing like $\sim x^{-1-\mu}$ and a typical value equal to $N^{-\gamma/2}$, we get

$$|G_{ii}^{(i)}|_{typ} \sim \frac{1}{1 + N^{-\gamma} N^{2/\mu} |G_{ii}^{(i)}|_{typ}}, \tag{B.2}$$

$$|G_{ij}^{(i)}|_{typ} \sim N^{-\gamma/2} |G_{ii}^{(i)}|_{typ}^2, \tag{B.3}$$

$$\text{and} \quad V_{ij} G_{jj}^{(i)} + \sum_{k \neq i,j} V_{ik} G_{kj}^{(i)} \sim N^{-\gamma/2} |G_{ii}^{(i)}|_{typ} + N^{-\gamma/2} N^{1/\mu} |G_{ij}^{(i)}|_{typ}. \tag{B.4}$$

Comparing the terms in the r.h.s. of the last equation, we get that the assumption is true for $\gamma\mu > 2$ and false otherwise.[13] Still, the diagonal approximation is sometimes used even for $\gamma\mu < 2$, as, e.g., in [31]. We do not know any analytical justification for that.

## C  Graphical algebra of (SFD-)symmetrically distributed r.v.s

Before trying to generalize the rules from Sec. 3 to the case of complex random variables, let us consider a more straightforward case, namely, the case of a real random variable supported on the entire real axis. The problem with this generalization is that $\alpha \in \mathbb{R}$ only describes an absolute value of the random variable $|x| = N^{-\alpha}$. Thus, ideally, we would need some additional construction to describe the sign-related aspects. For example, one could have written the PDF $p(x)$ of this random variable as $w^+ p^+(x) + w^- p^-(x)$, where $p^\zeta(x)$, $\zeta = \pm 1$, equals zero for $\zeta x < 0$ correspondingly, and assign conventional SFDs $f^\pm(\alpha)$ to each of the one-sided probability density functions $p^\pm(x)$ separately. However, it turns out that, for the purpose of the present paper, we only need to consider symmetric or, rather, SFD-symmetric distributions. Here, by SFD-symmetric, we mean that $f^+(\alpha) = f^-(\alpha) = f(\alpha)$ and $w^+ \propto w^- \propto N^0$, which is a weaker condition than $p(x) = p(-x)$.

After restricting ourselves to this case, we only need to adjust the meaning of $f(\alpha)$, its properties, and the rules discussed above. For example, while all even moments of symmetrically distributed random variables can still be found through their relation to tangent lines discussed around Fig. 1, all odd moments in terms of the symmetric SFD are now ill-defined. The exponentiation rule from Sec. 3.1 remains the same, with the only remark stating that even and odd powers now correspond to one-sided and SFD-symmetric distributions, respectively. The product rule from Sec. 3.4 and the ensemble-mixture rule from Sec. 3.3 stay exactly the same, and only the sum rule from Sec. 3.2 requires a substantial modification.[14] This modification is given below.

Consider two SFD-symmetric random variables $X$ and $Y$. Their sum is also SFD-symmetric and, hence, to draw $f_{X+Y}(\alpha)$, it is enough to know the probability density of the absolute value of the sum, which can be expressed as

$$p_{|X+Y|}(s) = (w_X^+ w_Y^+ + w_X^- w_Y^-) p_{|X|+|Y|}(s) + (w_X^+ w_Y^- + w_X^- w_Y^+) p_{|X|-|Y|}(s). \tag{C.1}$$

The SFD corresponding to $p_{|X|+|Y|}(s)$ can be calculated using the original sum rule from Sec. 3.2, the SFD corresponding to the weighted sum of different PDFs is given by the mix

---

[13]We base our conclusion on the assumption that the tail of the PDF of the off-diagonal contribution decays not slower than the diagonal contribution's one. Given the graphical algebra discussed above and the explicit exact expressions for the quantities involved, this assumption's validity is out of the question.

[14]The central limit theorem from Sec. 3.5 also needs to be modified. Moreover, for strictly symmetric, not only SFD-symmetric, distributions, the modification is sufficiently different from the original. However, since these modifications will not be of any use for us, we leave them as exercises for the reader.

rule from Sec. 3.3, so the only thing missing here is a rule for the subtraction of non-negative random variables. let us fill this gap. To do that, we write

$$p_{|X|-|Y|}(s) = w^+_{|X|-|Y|}p^+_{|X|-|Y|}(s) + w^-_{|X|-|Y|}p^-_{|X|-|Y|}(s), \tag{C.2}$$

where

$$p^\pm_{|X|-|Y|}(s) = \theta(\pm s)\int_{\max\{0,s\}}^\infty d\chi\, p_{|X|}(\chi)p_{|Y|}(\chi-s). \tag{C.3}$$

Then, we substitute $|s| = N^{-\alpha}$ and $\chi = N^{-\xi}$ and go from the PDFs to the SFDs:

$$p^\pm_{|X|-|Y|}(|s| = N^{-\alpha}) \propto \int_{-\infty}^\alpha d\xi\, N^{-\xi} N^{f_X(\xi)+\xi-1} N^{f_Y(\xi)+\xi-1} \propto N^{\max_{\xi\le\alpha}\{f_X(\xi)+f_Y(\xi)+\xi\}-2}. \tag{C.4}$$

This 'subtraction rule' is important by itself as it provides a way for further generalizations of the notion of the SFD to not only SFD-symmetric distributions. Finally, getting back to (C.1) and assuming, as earlier, that $\alpha_0(X) < \alpha_0(Y)$ and, for simplicity, that both $f_X(\alpha)$ and $f_Y(\alpha)$ are convex, we get

$$p_{X+Y}(|s| = N^{-\alpha}) \propto \begin{cases} N^{\max\{f_X(\alpha),f_Y(\alpha)\}+\alpha-1}, & \alpha < \alpha_0(X),\ \alpha_0(Y), \\ N^{\max_{\xi\le\alpha}\{f_X(\xi)+f_Y(\xi)+\xi\}-2} + N^{f_X(\alpha)+\alpha-1}, & \alpha_0(X) < \alpha < \alpha_0(Y), \\ N^{\max_{\xi\le\alpha}\{f_X(\xi)+f_Y(\xi)+\xi\}-2}, & \alpha > \alpha_0(X),\ \alpha_0(Y), \end{cases} \tag{C.5}$$

using the mix rule from Sec. 3.3 with equal weights $\propto N^0$. As one can see, the new contribution may prevent the suppression of zeros. This is, indeed, expected, considering that the zeros suppression effect originated from summing up strictly non-negative r.v.s. For example, adding together two normally distributed variables $X$ and $Y$ with $f_{X,Y}(\alpha) = (1-\alpha)\tilde\theta(\alpha)$, where $\tilde\theta(\alpha)$ is $-\infty$ for negative $\alpha$ and 1 for positive ones, we get back another normally distributed one as we should:

$$f_{X+Y}(\alpha) = \begin{cases} \max_{\xi\le\alpha}\{2f(\xi)+\xi\} - \alpha - 1 = 1 - \alpha, & \alpha \ge 0, \\ -\infty, & \alpha < 0. \end{cases} \tag{C.6}$$

Here, we used (1), given as $p_{X+Y}(|s| = N^{-\alpha}) \equiv N^{f_{X+Y}(\alpha)+\alpha-1}$.

# D  Graphical algebra of complex-valued Green's functions' SFDs

As one may guess, the task of generalizing the notion of the spectra of fractal dimensions to complex random variables is quite challenging. However, since we do not pursue the goal of being general but the goal of applying the algebra to our cavity Green's functions, we will generalize only those operations we actually need. let us see what the operations are:

1. Sum of independent real and complex random variables. This operation is necessary to compute the denominator of the r.h.s. of (20) and add together the real on-site energy and the complex self-energy.

2. Inverse of a random complex variable. This operation helps to get the Green's function from the sum of the on-site energy and the self-energy.

3. Product of independent real and complex random variables. This one is needed to compute individual terms of the sum corresponding to the (diagonal) cavity self-energy.

4. Sum of the extensive number of i.i.d. complex random variables (c.r.v.s). This is to compute the (diagonal) self-energy from the cavity Green's function and hopping amplitudes.

5. Taking real/imaginary parts of the c.r.v.s. This one is needed to finally get the distribution of the desired LDOS from the distribution of the complex-valued Green's function.

Restricting our interest to this set of operations, let us now define what we will mean by the SFD of a complex-valued random Green's function. The observation allowing us to simplify the definition significantly is the fact that, in the middle of the spectrum, the real part of Green's function is distributed (SFD-)symmetrically, see Appendix C for the definition of SFD-symmetric functions. Because of that, and because the imaginary part always has a one-sided distribution, we can define the desired SFD $f_X(\alpha_R, \alpha_I)$ such that $N^{f_X(\alpha_R,\alpha_I)+\alpha_R+\alpha_I-1}$ denotes a joint probability density for $X$ to have $|\operatorname{Re}X| \propto N^{-\alpha_R}$ and $|\operatorname{Im}X| \propto N^{-\alpha_I}$. The formal relation analogous to (1) is

$$
\begin{aligned}
p(|x_R|,|x_I|)\mathrm{d}x_R\mathrm{d}x_I &= p(N^{-\alpha_R},N^{-\alpha_I})N^{-\alpha_R-\alpha_I}\ln(N)^2\mathrm{d}\alpha_R\mathrm{d}\alpha_I \\
&= N^{f_N(\alpha_R,\alpha_I)-1}\ln(N)^2\mathrm{d}\alpha_R\mathrm{d}\alpha_I \, .
\end{aligned}
\tag{D.1}
$$

Thus, if $\operatorname{Re}X$ and $\operatorname{Im}X$ are in fact independent, $f_X(\alpha_R,\alpha_I) = f_{\operatorname{Re}X}(\alpha_R) + f_{\operatorname{Im}X}(\alpha_I) - 1$.

Because this generalization forces us to consider not planar diagrams but 2D surfaces embedded into 3D space, it requires some time to get used to the notation and find solid ground. To speed up this process, consider a couple of examples.

**2D Gaussian distribution** By definition, the probability density of the two-dimensional Gaussian distribution is a product of two independent one-dimensional PDFs. Assuming the first one has width $N^{-\alpha_{R0}}$ and the second one $N^{-\alpha_{I0}}$, we get the following SFD expression for the corresponding complex distribution:

$$
f_X(\alpha_R,\alpha_I) = (1-(\alpha_R-\alpha_{R0}))\tilde{\theta}(\alpha_R-\alpha_{R0}) + (1-(\alpha_I-\alpha_{I0}))\tilde{\theta}(\alpha_I-\alpha_{I0}) - 1 \, .
\tag{D.2}
$$

Here, as earlier, $\tilde{\theta}(x)$ is $-\infty$ for negative and 1 for positive values.

Picturing 2D surfaces on 2D paper clearly and concisely is generally challenging. To do that, we use polygon mesh: assuming our surfaces to be polyhedra and to have no curved regions, we plot different facets with different colors, mark edges and vertices with straight lines, arrows, and points, and specify edges' slopes with text labels. The 2D Gaussian distribution we defined above is depicted in Fig. 13a. While the contour plot of the surface would be more general and could depict both curved and polyhedral surfaces, it leads to, in some sense, a "raster" image requiring the specification of as many contours as possible for the complete representation of the surface. In contrast, the notation we propose relies, of course, on the absence of multifractality, but can describe each facet of a fractal SFD by just two vectors. In addition, it appears to be very convenient from the practical point of view: the aforementioned operations in this notation look easier than in any other we considered.

**Fully-correlated 2D Gaussian distribution** Next, let us see how correlations between our random variable's real and imaginary parts affect the picture. For simplicity, consider the extreme case with $\operatorname{Re}X = \operatorname{Im}X$ resulting in the Fig. 13b. The behavior along the only edge, which is shown as both $-\alpha_R$ and $-\alpha_I$, should be understood as follows: the labels are the coordinate-dependent parts of the parameterization of the SFD value along the edge, and, since it can be parameterized using both $\alpha_R$ and $\alpha_I$, both ways of labeling are possible and equivalent. In this particular case of the edge directed at $\pi/4$-angle to the axes, the change of the parameterization is particularly simple and leaves the functional dependence intact.

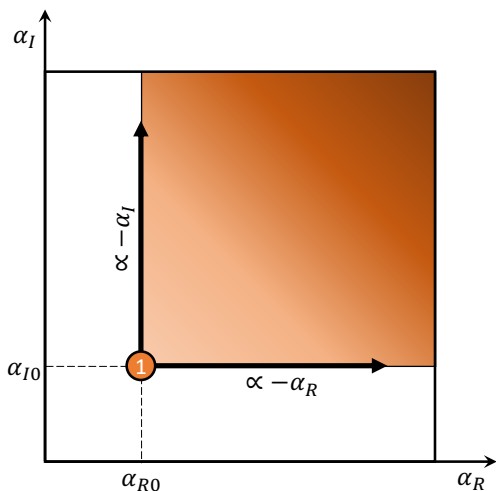

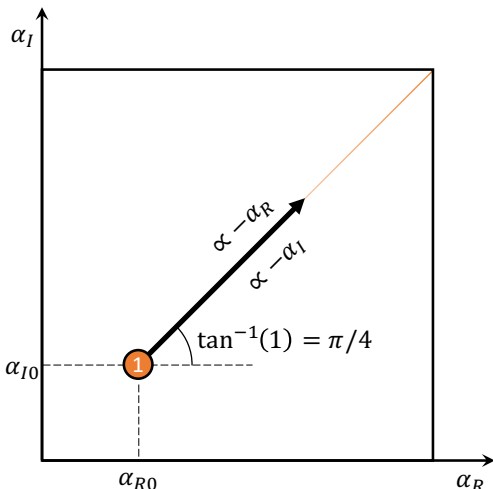

(a) The SFD (D.2) of a c.r.v. with real and imaginary parts being independent Gaussian r.v.s.

(b) The SFD of a complex variable $X + iXN^{\alpha_{R_0} - \alpha_{I_0}}$ with normally distributed $X$.

Figure 13: Graphical representations of the SFDs of c.r.v.s with (a) uncorrelated and (b) fully correlated real and imaginary parts. The solid point denotes the location of the typical (absolute) value of the random variable, while the encircled number gives the corresponding SFD value. The labels $\propto -\alpha_R$ and $\propto -\alpha_I$ show how the SFD behaves along the edges. Notice that the diagonal edge on panel (b) has both labels close to it. This is only to demonstrate their equivalence as any edge having non-zero angles to both axes can be labeled using both $\alpha_R$ and $\alpha_I$, up to the author's choice. Below, we will choose one of the two ways of labeling for each specific situation separately. The colorful zone represents a facet with a finite value of the SFD, while the white zone represents the area with $f(\alpha_R, \alpha_I) = -\infty$. In addition, the arrows show the direction of the SFD's decay along the edges. This does not provide any additional information but makes it easier to read the diagrams. The value of the SFD on the facet is schematically shown by the color gradient and is only present here for the aesthetic beauty.

## D.1 Real and imaginary parts of the complex r.v.s.

Imagine we have obtained the SFD for our complex-valued Green's function. How to extract the information about the local density of states from it? To do that, we need to calculate the SFD $f_{\mathrm{Im}X}(\alpha)$ of the marginal distribution of the Green's function's imaginary part. From the definitions (1) and (D.1), we find that

$$N^{f_{\mathrm{Im}X}(\alpha)} \propto \int d\alpha_R N^{f_X(\alpha_R, \alpha_I)} \propto N^{\max_{\alpha_R}\{f_X(\alpha_R, \alpha_I)\}} \implies f_{\mathrm{Im}X}(\alpha_I) = \max_{\alpha_R}\{f_X(\alpha_R, \alpha_I)\},$$
(D.3)

the expression for $f_{\mathrm{Re}X}(\alpha)$ is analogous. Pictorially, drawing a marginal SFD from the joint SFD is an orthographic projection along the corresponding axis, see, e.g., Fig. 14.

## D.2 Sum of the real and complex r.v.s.

A PDF $p_X(x_I, x_R)$ of any distribution can be written as $p_{\mathrm{Im}X}(x_I)p_{\mathrm{Re}X}(x_R|x_I)$, where $p_{\mathrm{Im}X}(x_I)$ is a PDF of the marginal distribution of $\mathrm{Im}X$ and $p_{\mathrm{Re}X}(x_R|x_I)$ is a PDF of the conditional distribution of $\mathrm{Re}X$ given that $\mathrm{Im}X = x_I$. Noticing that r.v. $X + Y$ has the same $p_{\mathrm{Im}X}(x_I)$

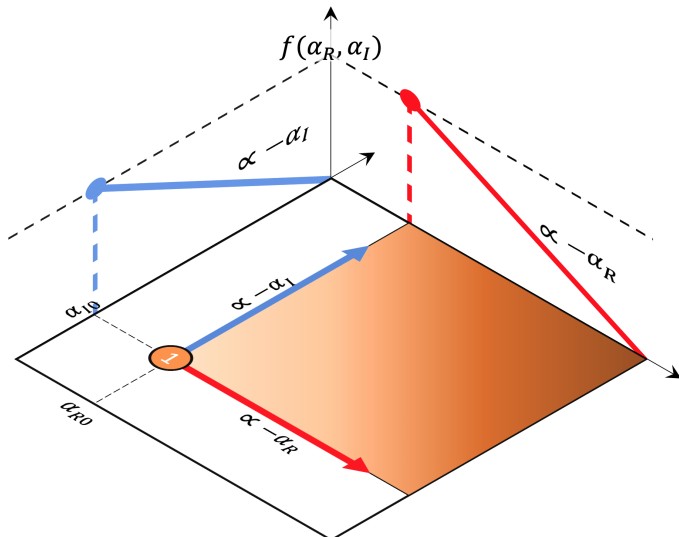

Figure 14: Orthographic projections of the SFD from Fig. 13a. The colors of the projections indicate edges contributing to these projections. Notice that the projections would be the same also for the SFD from Fig. 13b.

as $X$ provided $Y \in \mathbb{R}$, we reduce the task of calculating the distribution of $X + Y$ to the task of calculating the distributions of $Y + \operatorname{Re} X$ for different fixed values of $\operatorname{Im} X$ which can be done using the already familiar rules from Sec. 3.2 and Appendix C. Thus, the corresponding graphical algorithm is:

1. Split the 2D surface $f_X(\alpha_R, \alpha_I)$ of the r.v. $X$'s SFD into individual 1d slices with fixed $\alpha_I$.

2. Shift each of the slices vertically such that their maxima become 1. After this normalization, the slices can be viewed as the SFDs $f_{\operatorname{Re} X}(\alpha_R|\alpha_I)$ of the corresponding conditional distributions.

3. Calculate the SFD of the sum $\operatorname{Re} X + Y$ under the condition, posed by each of these slices, using the summation and subtraction rules from Sec. 3.2 and Appendix C.

4. Assemble the 2D surface from the resulting 1d slices restoring their initial maximal heights by the appropriate vertical shifts given by the SFD $f_{\operatorname{Im} X}(\alpha)$.

To illustrate the scheme, consider a sum of the complex fully correlated Gaussian r.v. $X$ from Fig. 13b and the real one-sided Lévy r.v. $Y$ from (37) with $c > \alpha_{R0}$. The normalized slices with fixed imaginary $\alpha_I$, $f_{\operatorname{Re} X}(\alpha_R|\alpha_I)$, of the two-dimensional SFD of $X$ together with the corresponding vertical shifts given by the marginal SFD $f_{\operatorname{Im} X}(\alpha_I)$ are

$$
f_{\operatorname{Re} X}(\alpha_R|\alpha_I): \qquad\qquad\qquad f_{\operatorname{Im} X}(\alpha_I): \qquad\qquad \propto -\alpha_I \qquad (\text{D.4})
$$
$$
\alpha_{R0} + (\alpha_I - \alpha_{I0}) \qquad\qquad\qquad\qquad \alpha_{I0}
$$

Notice that the slices are associated with symmetric conditional distributions, but the results of Sec. C cannot be applied directly because the Lévy distribution is not symmetric. However, keeping in mind (C.4), the remark directly after it, and considering the SFD of $X + Y$ as the equal-weight mix of a sum and a difference between the Lévy and the half-normal distributions, we get the following result[15] for the $f_{\operatorname{Re} X}(\alpha_R|\alpha_I)$ of the individual slices for SFD of the sum

---

[15]Notice that the result for $\alpha_0(\alpha_I) < c$ would be different if this was a sum of two one-sided distributions. Indeed, the suppression of zeros would lead to $f(\alpha > c) = -\infty$.

$X + Y:$

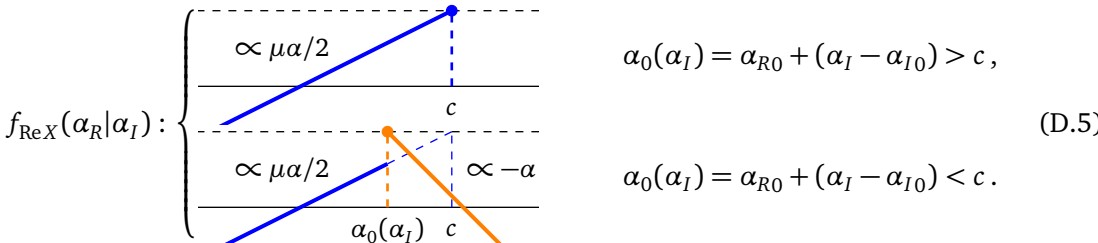

$$f_{\mathrm{Re}X}(\alpha_R|\alpha_I):\begin{cases} & \alpha_0(\alpha_I) = \alpha_{R0} + (\alpha_I - \alpha_{I0}) > c, \\ & \\ & \alpha_0(\alpha_I) = \alpha_{R0} + (\alpha_I - \alpha_{I0}) < c. \end{cases} \tag{D.5}$$

Finally, assembling all the slices together with the heights defined by $f_{\mathrm{Im}X}(\alpha_I)$, we get

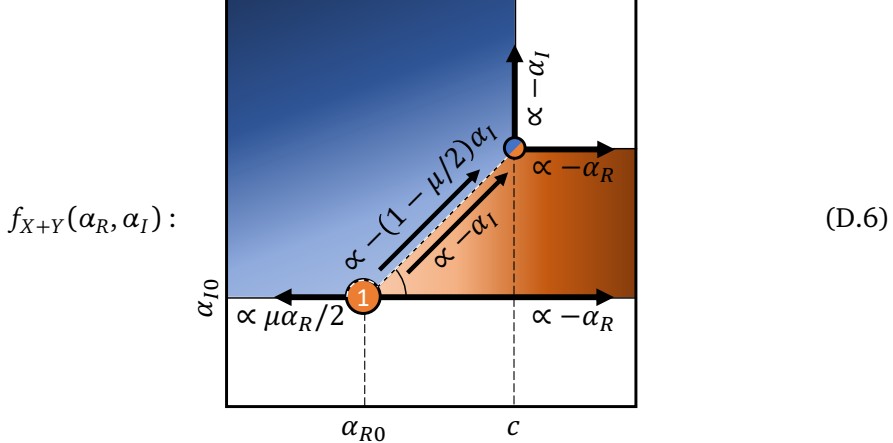

$$f_{X+Y}(\alpha_R, \alpha_I): \tag{D.6}$$

Here and further, we omit the specification of vertical and horizontal axes, assuming them always to be $\alpha_R$ and $\alpha_I$. For brevity, we also omit the specification of the $\pi/4$-angle. From now on, we will always use the same solid-line arc notation to specify such angles. The different colors of the facets mark their relation to the original r.v.s $X$ and $Y$, allowing, as earlier, to trace back easily the features we see in the resulting distribution. The black-and-white dashed line connecting the vertices marks the discontinuity between the facets originating from the second case in (D.5), and the colors of the vertices mark the facets they belong to: the vertex at $\{\alpha_{R_0}, \alpha_{I_0}\}$ belongs only to the orange facet, while the other vertex belongs to both. Consequently, the two arrows and two non-equivalent labels below and above the discontinuous edge show the slopes for the orange and the blue facets, respectively.

### D.3   Product of the real and complex r.v.s

To start with, consider a generic complex r.v. $X$ and multiply it by a constant $C = N^{-c}$. The resulting two-dimensional SFD of $CX$ is just a shift of the original one: $f_{CX}(\alpha_R, \alpha_I) = f_X(\alpha_R + c, \alpha_I + c)$. A product of $X$ and a generic real random variable $Y$ can be viewed as a weighted ensemble mixture of such shifts, with the weights controlled by the SFD of $Y$. Moreover, since all the shifts happen along the lines $\alpha_R - \alpha_I = s = \mathrm{const}$, one can view the product operation similarly to the previously considered sum operation, i.e., as applying the same independent real operations to different slices characterized by different values of $s$.

To better understand the operation, consider a product of $X$ sampled from the uncorrelated 2D Gaussian distribution from Fig. 13a and $Y$ sampled, again, from the Lévy distribution (37). For $s < s_0 = \alpha_{R_0} - \alpha_{I_0}$, the normalized fixed-$s$ slices of the complex SFD of $X$, parameterized by $\alpha_R$, together with the Lévy distribution's SFD, look like

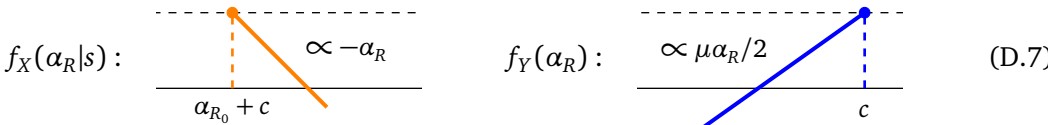

$$f_X(\alpha_R|s): \qquad\qquad f_Y(\alpha_R): \tag{D.7}$$

for $s > s_0$, the diagrams are the same up to the substitution $R \to I$. Then, by multiplying these two SFDs according to the usual rules from Sec. 3.4 and assembling different slices together according to the corresponding weights of each slice, we get

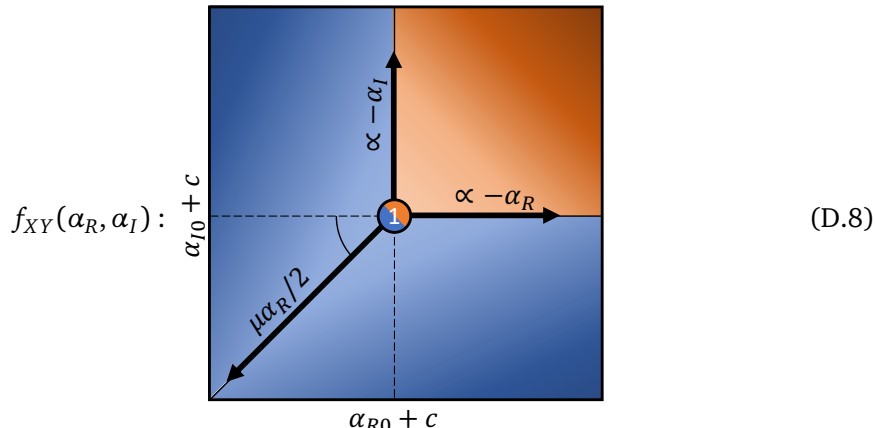

$$f_{XY}(\alpha_R, \alpha_I): \qquad\qquad\qquad\qquad\qquad\qquad\qquad\qquad\qquad (D.8)$$

The correctness of this result can be again checked by multiplying $Y$ separately by (independent) $\mathrm{Re}\,X$ and $\mathrm{Im}\,X$ and assembling the complex-valued r.v. from its (independent) real and imaginary parts.

## D.4 Inversion of the complex random variable

In general, the graphical exponentiation of complex r.v.s is a mess because of the mixing of different phases. However, to calculate cavity Green's function, we only need to learn how to get an SFD of $1/X$ from the SFD of $X$, which is doable. The rule for the inversion operation can be deduced from the following two observations:

1. Inversion preserves the argument of c.r.v. mod $\pi$. This means that if the point of the original 2D plane corresponded to the tangent $N^{-\alpha_I}/N^{-\alpha_R}$, its image will correspond to the same tangent. In other words, the quantity $\alpha_I - \alpha_R$ conserves for all points under the inversion operation. This leads to the conclusion that all actions again occur along the $\pi/4$ slices, as with the product discussed above.

2. Inversion operation inverses the absolute value, which is, in our case, dominated either by $N^{-\alpha_R}$ or $N^{-\alpha_I}$. Thus, if the original point lies above the line $\alpha_I = \alpha_R$, its $\alpha_R$ changes the sign. The same happens with the $\alpha_I$ for the points below the line.

The resulting recipe for drawing the SFD of a complex random variable's inverse is shown in Fig. 15.

In Fig. 15b, we illustrated how the angles $\arctan(2)$ and $\arctan(1/2)$ may arise on the diagrams from the inversion operation. Because they will often appear in real calculation, we introduce a special notation for them: we specify the former with a double-line arc and the latter with a dashed-line arc. Below, we omit the text labels and only use this notation.

## D.5 Generalized central limit theorem for c.r.v.s

Finally, let us derive the rule for extensive summation of i.i.d. complex r.v.s. For simplicity, we do it in a graphical fashion similarly to how we did it in Sec. 3.5, and not in a mathematical style of Appendix A. Thus, we reevaluate the arguments surrounding (16). What this equation says is that each set of terms $x_i \propto N^{-\alpha_i}$, considered individually, sums up to a definite quantity of the order of $N^{-\alpha_i} N^{f_X(\alpha_i)+1-\beta} \delta\alpha$, where $N^{f_X(\alpha)+1-\beta} \delta\alpha$ is the expected number of such terms. The final result is obtained then as a sum of all such contributions from different $\alpha$ and equals

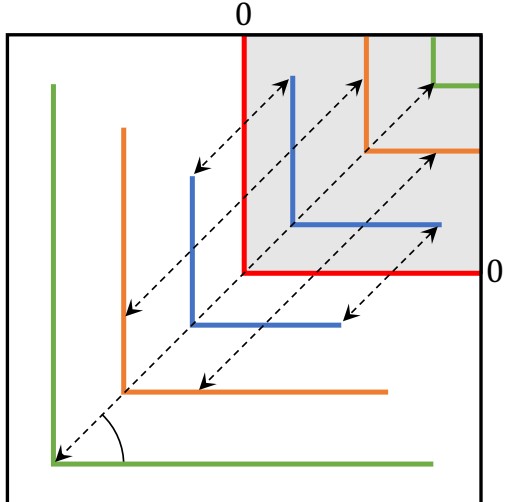

(a) Inversion-invariant shapes, origins and images are shown by the same color.

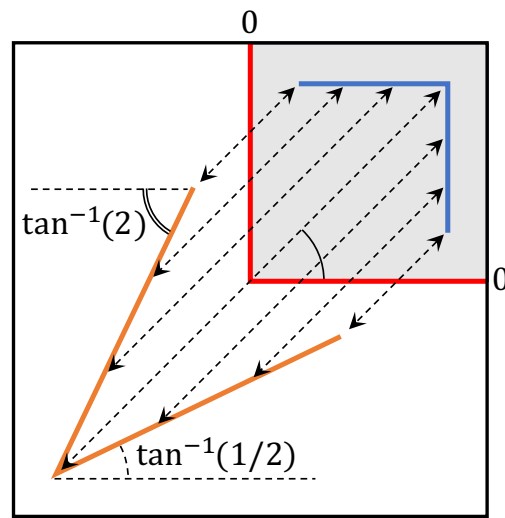

(b) A general case of inversion, an origin and an image are shown by different colors.

Figure 15: Graphical representations of the behavior of the SFD of a c.r.v. under inversion. The points connected by the double-sided arrows are the source and the image of each other. The red line shows a set of points invariant under the inversion operation. Therefore this red line separates each of the dashed double-sided arrows into two equal parts. The white and grey sectors of the complex plane are the source and the image of each other.

to the dominant one coming from $\alpha_i$ such that $f'_X(\alpha_i) = 1$ (saddle point). Now, we are going to use exactly the same logic here: our starting point for the derivation is the analysis of extensive sums of r.v.s defined by narrow SFDs concentrated around a single point on the $\{\alpha_R, \alpha_I\}$ plane. In the case of the one-sided distribution with an SFD concentrated around $\alpha = \gamma$, an SFD of a sum of $N^\beta$ such i.i.d. random variables would be just another similar SFD concentrated around $\gamma - \beta$. In the case of general SFD-symmetric distribution, the result would be equal to a subtraction of the absolute values of the extensive sums of positive and negative terms, leading to an SFD with the same typical amplitude[16] but also with the linear part $\propto -\alpha$ to the right of $\gamma - \beta$. In the case we are discussing now, the answer would depend on the nature of real and imaginary parts of our complex random variable.

### D.5.1 One-quarter complex distribution.

First, assume both real and imaginary parts of our random summand correspond to one-sided distributions. This situation results in the following expression:

$$\sum_{i=1}^{N^\beta} \boxed{\begin{array}{c} \alpha_{I0} \\ \cdots \bullet 1 \\ \alpha_{R0} \end{array}} = \boxed{\begin{array}{c} \alpha_{I0} - \beta \\ \cdots \bullet 1 \\ \alpha_{R0} - \beta \end{array}} \tag{D.9}$$

Indeed, since, in this case, the sum of identical terms is just a multiplication by $N^\beta$, it leads to a simple shift of the initial concentrated SFD. And, since the shift is again happening solely

---

[16]As we have already mentioned above in the footnote 14, the case of strictly symmetric distributions is qualitatively different from the more general SFD-symmetric one. Nevertheless, the results of this section can be generalized to this special case using the same strategy.

along the lines with $\alpha_R - \alpha_I = s =$ const, we can, in principle, treat the extensive sum of i.i.d. c.r.v.s similarly to how we approached the product and the inverse, i.e., by applying the corresponding real operation to individual s-slices and *properly* assembling the results together.

However, the rules for this 'proper assembling' are quite tricky, and one should be careful while summing up the contributions from different slices. To see why, consider an SFD with not one but two peaks. If they correspond to different values of $s$ and, hence, do not lie on the same slice, the resulting extensive sum may look like

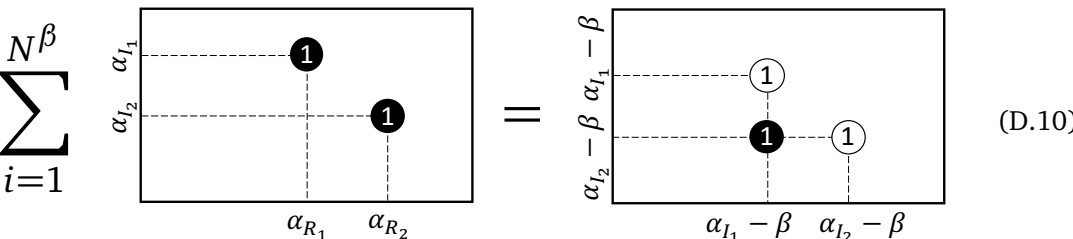

$$(D.10)$$

On the r.h.s. diagram, the white circles show separate contributions from each peak, and the black circle shows the actual typical value of the sum. Notice that the real and imaginary parts of this value come from different peaks' contributions: its $\alpha_R$ is equal to $\min\{\alpha_{R_{0_1}}, \alpha_{R_{0_2}}\} - \beta$, and its $\alpha_I$ is equal to $\min\{\alpha_{I_{0_1}}, \alpha_{I_{0_2}}\} - \beta$. In other words, the sum has the same binary nature as the sum of two non-negative r.v.s from Sec. 3.2. This nature is most evident from the expression

$$\left(N^{\beta-\alpha_{R_{0_1}}} + iN^{\beta-\alpha_{I_{0_1}}}\right) + \left(N^{\beta-\alpha_{R_{0_2}}} + iN^{\beta-\alpha_{I_{0_2}}}\right) = N^{\beta}\left(\left(N^{-\alpha_{R_{0_1}}} + N^{-\alpha_{R_{0_2}}}\right) + i\left(N^{-\alpha_{I_{0_1}}} + N^{-\alpha_{I_{0_2}}}\right)\right)$$
$$\propto N^{\beta}\left(N^{-\min\{\alpha_{R_{0_1}}, \alpha_{R_{0_2}}\}} + iN^{-\min\{\alpha_{I_{0_1}}, \alpha_{I_{0_2}}\}}\right).$$
$$(D.11)$$

For summands with SFDs having more than two finite-valued slices, the generalization is similar to the one for the sum rule from Sec. 3.2 to any $N$-independent number of terms. In other words, using the same coarse-graining argument for the slices-enumerating parameter $s$ as we exploited in Appendix A for $\alpha$, we arrive at the conclusion that *all extensive sums of i.i.d. c.r.v.s must have SFDs topologically similar to*

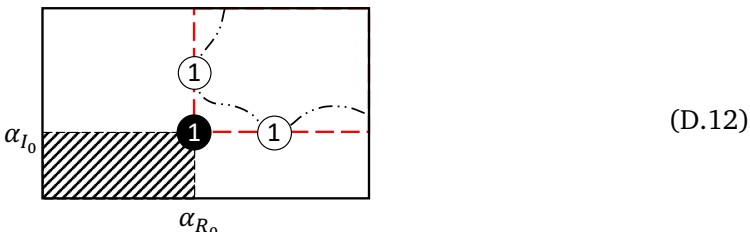

$$(D.12)$$

here, the double-dot-dashed line $\{\alpha_R(s), \alpha_I(s)\}$ represents the manifold of typical contributions from different s-slices, and the right angle shown by the red dashed line is fixed by the two points on the manifold, the ones with $\alpha_{R,I_0} = \min_s \alpha_{R,I}(s)$. Consequently, the black point $\{\alpha_{R_0}, \alpha_{I_0}\}$ represents a typical value of the corresponding extensive sum, and the rest of the finite-valued sum's SFD, if any, must lie in a dashed quarter-plane region. This is what the zeros suppression effect looks like on the complex plane.

Now, after clarifying the situation with the typical value of the extensive sum, let us find out what happens to the distribution's tails, i.e., how rare events contribute to the sum, and how the dashed quarter-plane region from (D.12) must look like in any particular case. For this purpose, consider i.i.d. summands $X$ with the SFD $f_X(\alpha_R, \alpha_I)$, an arbitrary point $\{\alpha_R, \alpha_I\}$ from the dashed region of (D.12), and calculate the probability for the extensive sum $S$ to take the corresponding value. In contrast to the real-valued case from Sec. 3.5 when the only significant

contribution to $f_S(\alpha < \alpha_0)$ came from rare terms of the order of $N^{-\alpha}$, in the complex-valued situation, there is another worth-mentioning possibility to get the extensive sum's value of the order of $N^{-\alpha_R} + iN^{-\alpha_I}$: namely, apart from having one very rare term of this order, one may consider having two less rare terms of the orders $N^{-\alpha_R} + iN^{-\tilde{\alpha}_I}$ and $N^{-\tilde{\alpha}_R} + iN^{-\alpha_I}$, with $\tilde{\alpha}_{R,I} > \alpha_{R,I}$. As a result, the correct expression describing the tails of the SFD $f_S(\alpha_R, \alpha_I)$ is

$$f_S(\alpha_I < \alpha_{I_0}, \alpha_R < \alpha_{R_0}) = \max\left\{f_X(\alpha_R, \alpha_I), \max_{\tilde{\alpha}_I > \alpha_I}\{f_X(\alpha_R, \tilde{\alpha}_I)\} + \max_{\tilde{\alpha}_R > \alpha_R}\{f_X(\tilde{\alpha}_R, \alpha_I)\} - 1\right\} + \beta\,. \tag{D.13}$$

The first option under the outer MAX operation in the r.h.s represents the direct contribution from the point $\{\alpha_R, \alpha_I\}$, while the second option represents the aforementioned two-point contribution. As the terms corresponding to the points are independent, the probability density for them to occur in the same sum is a product of their individual probability densities, hence the sum of the corresponding SFDs minus one.

As one can guess, the outer MAX operation in (D.13) may result in an additional facet on the sum's SFD compared to the SFD of the summand. An example of this behavior is shown in

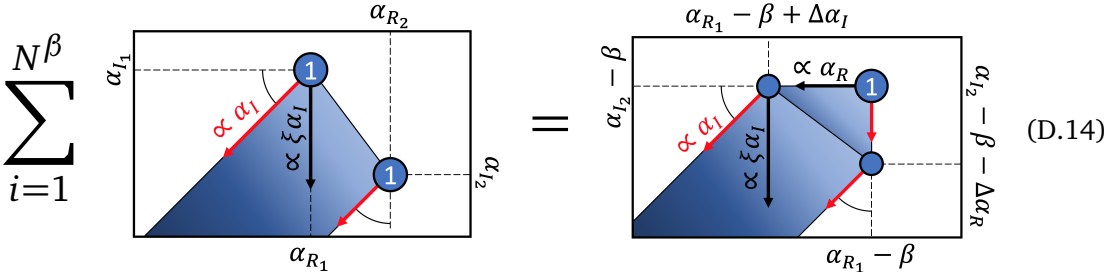

$$\tag{D.14}$$

Here, $\Delta\alpha_{R,I} = \alpha_{R,I_2} - \alpha_{R,I_1}$, $\xi = \Delta\alpha_R/(\Delta\alpha_R + \Delta\alpha_I)$ is a constant consistent with the zero slope between the encircled 'ones' at the points $\{\alpha_{R_i}, \alpha_{I_i}\}$, and, for brevity, we assume all colored arrows of the same color within the same diagram to have the same labels. This latter convention appears to be very useful on dense diagrams with many different facets. A detailed diagram corresponding to the same sum is given by

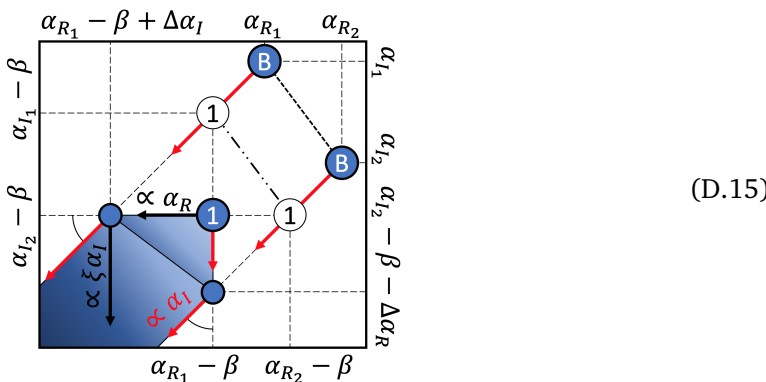

$$\tag{D.15}$$

Here, the upper-right blue points with the heights $B = 1 + \beta$ and the small dashed line in-between them schematically represent the SFD of the summand $X$ elevated by the value of $\beta$ as we usually pictured at similar diagrams corresponding to the real-valued random variables, see Fig. 6. Continuing the analogy, we used the double-dot-dashed line to mark the points where the slice-wise unit-slope lines, having exactly one common point with $f_X(\alpha_R, \alpha_I) + \beta$ in the area where $f_X(\alpha_R, \alpha_I) + \beta \geq 1$, cross the plane $f(\alpha_R, \alpha_I) = 1$. As one can see, the positions and heights of the resulting SFD's vertices then follow from this geometrical construction.

Thus, for the extensive sum of c.r.v.s with a generic PDF different from zero only in a single quarter of the complex plane, the graphical rules can be formulated as follows:

1. For each slice $s = \alpha_R - \alpha_I$ parameterized by either $\alpha_R$ or $\alpha_I$, apply the extensive summation rule according to the algorithm from Sec. 3.5.

2. Draw the curve $\{\alpha_R(s), \alpha_I(s)\}$ corresponding to the typical contributions from each slice and determine the position of the typical value of the whole sum as $\{\min_s \alpha_R(s), \min_s \alpha_I(s)\}$.

3. Complete the tails of the SFD with the use of (D.13).

### D.5.2 Two-quarter complex distribution.

A case when the PDF of a distribution is non-zero not in one but in two quarters of the complex plane is the most relevant for our physical application. Indeed, since the associated with the local density of states imaginary part of Green's function is always greater than or equal to zero while its real part in the bulk has an SFD-symmetric distribution, the distribution of the complex Green's function falls exactly into the category of two-quarter complex distributions.

So now, without loss of generality, assume the half-plane corresponding to a non-zero PDF to be the upper half-plane. Then, as at the beginning of App. D.5.1, consider the trial SFD concentrated around a single value of $\{\alpha_R, \alpha_I\}$, meaning that all of the terms in the sum are assumed to have the real and imaginary parts of the order of $N^{-\alpha_{R_0}}$ and $N^{-\alpha_{I_0}}$ correspondingly. Since the imaginary part of our trial random variable is non-negative, the imaginary part of the sum is also non-negative, and the zeros suppression effect for the imaginary part takes place as usual, meaning that the SFD of the extensive sum of $N^\beta$ such terms cannot have finite values away from the line $\alpha_I = \alpha_{I_0} - \beta$. However, since the real part of the individual terms is assumed to be SFD-symmetric, the real part of the sum does not show the same zeros suppression. To see this, it is enough to separate the sum's terms into two partial sums consisting of the terms with positive and negative real parts, sum up each of the partial sums separately, and subtract one from another according to (C.4). As a result, instead of (D.9), we arrive at

$$\sum_{i=1}^{N^\beta} \pm \; \boxed{\substack{\alpha_{I0} \\ \bullet 1 \\ \alpha_{R0}}} \;=\; \boxed{\substack{\alpha_{I0}-\beta \\ \bullet 1 \; {\propto\, -\alpha_R} \\ \alpha_{R0}-\beta}} \tag{D.16}$$

Analogously, after the same partitioning of the sum, instead of (D.10), we obtain another diagram with the same position of the peak but with the additional linear part $\propto -\alpha_R$ to the right of it. Finally, for the extensive sum of i.i.d. c.r.v.s having a generic two-quarter distribution, after the same partitioning and summing up the partial sums according to App. D.5.1, the result can be obtained as a mix of all contributions from all possible pairs of points corresponding to the two partial sums, meaning that the only modification we need to introduce to the rule from App. D.5.1 is the addition of the linear parts $\propto -\alpha_R$ to the right of each point. As an example, the extensive sum from (D.14) but with an SFD-symmetrized[17] real part would look

---

[17] As we already mentioned in the footnotes 14 and 16, the case of strictly symmetrized terms is qualitatively different!

like

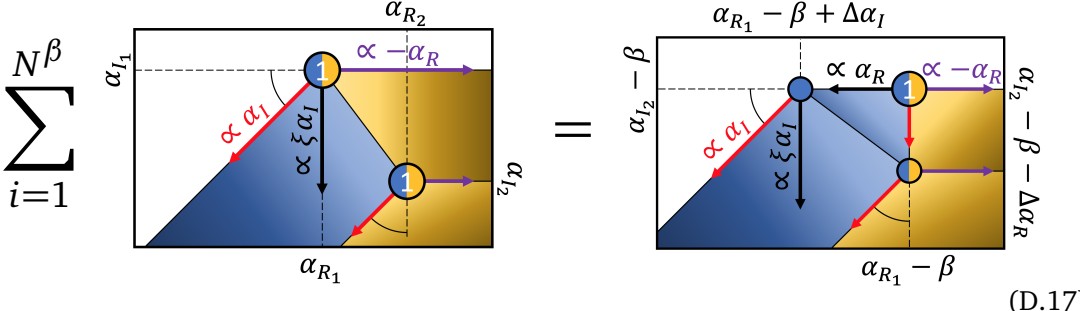

$$\sum_{i=1}^{N^\beta} \quad = \qquad (D.17)$$

Here we use the same notion of $\xi$ from (D.14).

The case of the distributions supported on the whole complex plane can be considered similar to the ones described above. However, since our applications do not touch this topic, we will leave this generalization as an exercise for the reader.

### D.6 Full Gaussian RP cavity LDOS SFD calculation

Now, after developing all the necessary techniques, let us try to self-consistently calculate a full SFD of the Gaussian RP Green's function. As we usually did in the cases of one-sided real LDOS distributions, let us start from an educated guess. In this case, let us guess the cavity self-energy $\Sigma_i = \sum_{j \neq i} |V_{ij}|^2 G_{jj}$, with $\Sigma_{typ} \propto N^{-c_R} + iN^{-c_I}$. What we know about this quantity is that, due to the absence of the tails in its distribution and the zeros suppression effect, its imaginary part has an SFD concentrated around $N^{-c_I}$ (let us pretend we do not know that $c_I = \gamma - 1$). We can also assume that, since $\mathrm{Re}\,\Sigma_i$ in the middle of the spectrum definitely has a non-zero finite PDF at $\mathrm{Re}\,\Sigma_i = 0$, its marginal SFD is of the Gaussian type, like (23) or (C.6). These observations, after considering the rule depicted in Fig. 14, result in the guess

$$\Sigma: \qquad (D.18)$$

The rest of the calculations can be summarized as follows. After adding the independent and normally distributed on-site disorder to the above self-energy according to App. D.4, we get

$$\varepsilon_i + \Sigma_i: \qquad (D.19)$$

Then, after inverting this result according to App. D.4, we get

$$G_{ii} = (\varepsilon_i + \Sigma_i)^{-1}: \qquad (D.20)$$

one can see how the fracture of the edge from (D.19) happens on the line $\alpha_R = \alpha_I$, in agreement with the rules pictured in Fig. 15. Also, in (D.20), we label the edge from $\{0, c_I\}$ to $\{-c_I, -c_I\}$ using the variable $\alpha_I$ to simplify taking the imaginary part of the Green's function, as, in this parameterization, it is immediately clear that, according to App. D.1, the LDOS SFD calculated as $\mathrm{Im}\, G_{ii}$ with $G_{ii}$ from (D.20) is equal to (27), as it should. Then, after multiplying (D.20) by $|V_{ij}|^2$ from (29) according to App. D.3, we obtain

$$G_{ii}|V_{ij}|^2 : \qquad \text{(D.21)}$$



where we changed the parameterization of the edge with a red arrow from $\alpha_I$ in (D.20) to $\alpha_R$ in (D.21) to simplify the drawing of the double-dot-dashed contour line in the final expression for the self-energy SFD

$$\Sigma_i = \sum_{j \neq i} G_{jj}|V_{ij}|^2 : \qquad \text{(D.22)}$$

In this parameterization, it becomes clear that the typical contributions from the whole edge to the extensive sum from (D.22) have the same real part, $\alpha_R = \gamma - 1$.

From this calculation, we see that the typical values of $\mathrm{Re}\,\Sigma_i$ and $\mathrm{Im}\,\Sigma_i$ have the same scaling and the result (D.20) gives the same LDOS as we obtained earlier in Sec. 4 neglecting $\mathrm{Im}\,\Sigma_i$. But probably the most useful observation here is the fact that the interior of the colored area from (D.21) had no impact on the distribution of the self-energy. This is, of course, just a trivial consequence of the complex version of the zeros suppression effect, but, still, it will have a significant consequence for some of our considerations in App. E. In addition, one can see a curious fact that the marginal SFD of the real part of the RP Green's function (D.20) is not trivial and reminds the SFD we obtained for the Lévy-RP in Sec. 5.

## E  Full Lévy-RP cavity LDOS SFD calculation

Finally, let us self-consistently calculate a full SFD of the Lévy-RP Green's function and, hopefully, obtain the desired local density of states in agreement with our numerical experiments, Fig. 8. However, instead of guessing the self-energy distribution for the Lévy-RP model, let us exploit a different approach: let us start from some arbitrary SFD and iterate the diagonal cavity Green's function expression (20) until the iterations converge. A choice of the same self-energy SFD as in the Gaussian RP case looks like a good option, so let us start from (D.18).

Then, we naturally arrive at (D.20) and, after multiplying it with $|V_{ij}|^2$ defined by

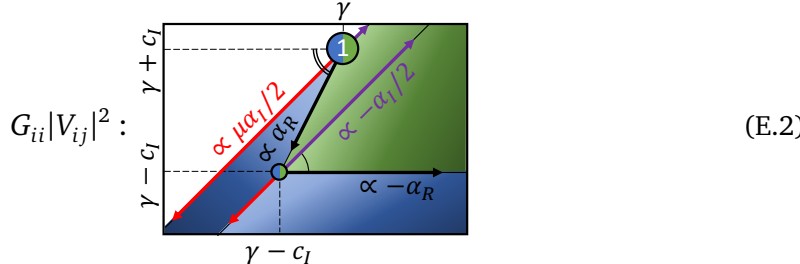

$$|V_{ij}|^2: \tag{E.1}$$

get

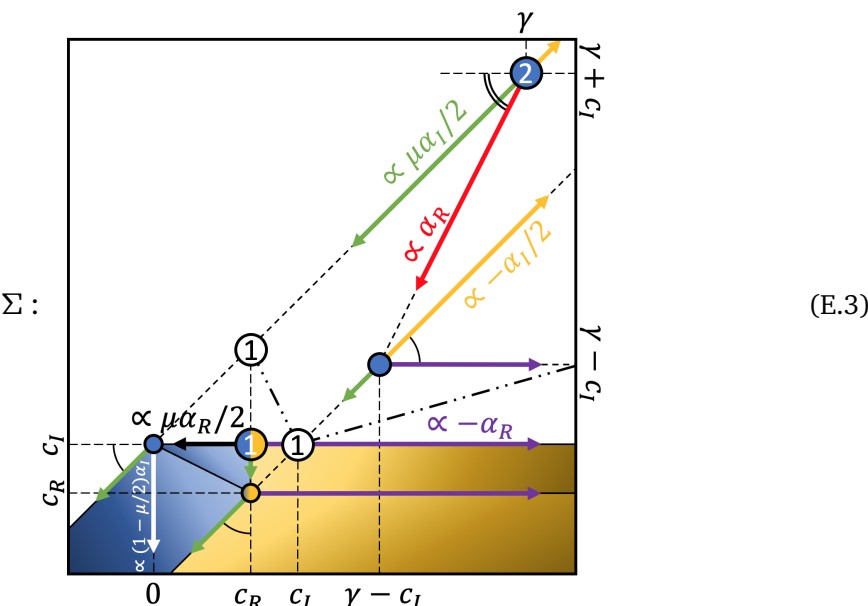

$$G_{ii}|V_{ij}|^2: \tag{E.2}$$

here, we used different colors for different facets to specify where these facets originated from with respect to (E.1). From what we got, one can already see that our initial guess for the self-energy was incorrect. However, this was a part of the plan, so let us continue. The second-iteration guess for the self-energy obtained from (E.2) looks like

$$\Sigma: \tag{E.3}$$

Here, the small triangular blue facet originates from the tails of (E.1) because of (D.13), and the yellow facets appear because of the SFD-symmetric nature of the Green's function's real part, see App. D.5.2. The absence of any green facets signals that, again, as in the Gaussian RP case considered in App. D.6, the (green) interior of the angle from (E.2) formed by the black arrows does not contribute to the self-energy distribution.

Continuing our iterations using (E.3) as the self-energy distribution, and adding it to the

diagonal disorder defined by (23) according to App. D.2, we get

$$\varepsilon_i + \Sigma_i :$$ 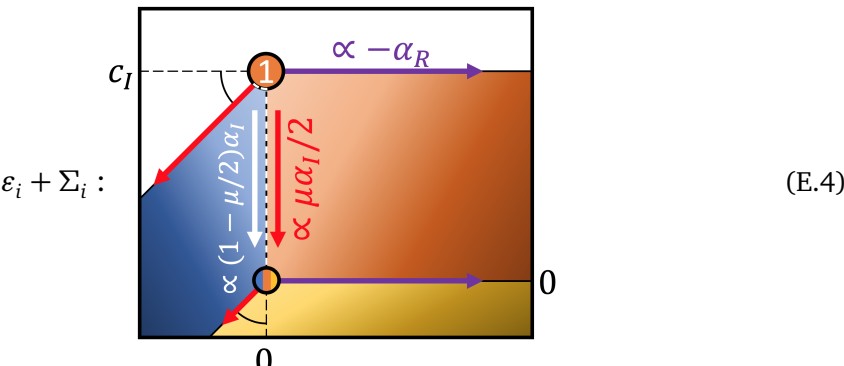 (E.4)

here, the blue and the yellow facets originate from the corresponding facets in (E.3), while the orange facet originates from the on-site energies distribution (23). Notice the black-and-white dashed line connecting the vertices $\{0, c_I\}$ and $\{0, 0\}$ and the filling colors of the vertices. As it was already introduced in App. D.2, this notation depicts discontinuity between the blue and orange facets.

Finally, by inverting (E.4) according to App. D.4, we get the complex Green's function

$$G_{ii} = (\varepsilon_i + \Sigma_i)^{-1} :$$ 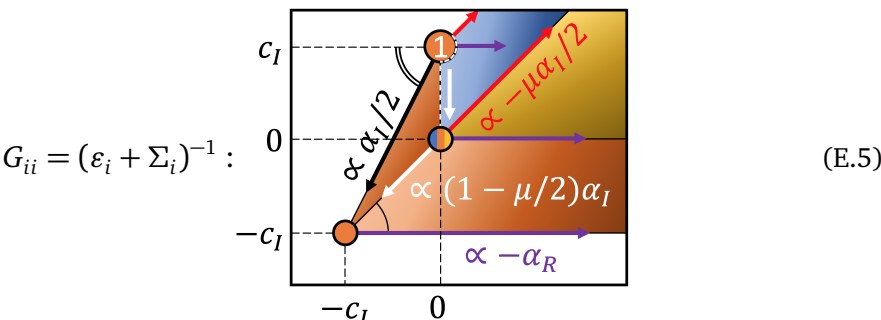 (E.5)

As one can see, the exterior of the colored area and the slopes of its borders are identical to those from the Gaussian RP (D.20). And, since, for $\mu > 1$, the slope of the white arrow parameterized by $\alpha_I$ is always smaller than $\mu/2$, the interior of the area will not contribute to the essential parts of the subsequent calculations, similar to how it happened in the previous cases. These two facts allow us to claim that the subsequent iterations with the use of Green's function $G_{ii}$ from (E.5) will lead to the exactly the same result as we obtained using (D.20). The SFD of $G_{ii}|V_{ij}|^2$ will differ from the result (E.2) only by the insignificant changes in the green area, and the self-energy will then fully coincide with the result (E.3), making our solution self-consistent, and the result (E.5) the desired answer. Thus, we can stop here and write the self-consistency equations based on (E.3). From that geometrical construction, one finds that $c_I = (\gamma - 2/\mu)/(2 - 2/\mu)$, in agreement with the value following from (40). In its turn, $c_R = \gamma - 2/\mu$ is not equal to and even smaller than $c_I$, in contrast to the Gaussian RP case and (D.22).

Finally, let us extract the real-valued LDOS SFD from this complex Green's function distribution (E.5). It can be done using the projection rule from App. D.1 that gives

$$\text{Im}\, G_{ii} = \nu_i :$$  (E.6)

here, the colors of the lines, again, reflect the colors of the facets whose edges contributed to the result, revealing its origins. The difference with the result (38) is in the value of $f_\nu(c_I + 0)$.

While previously we got that $f_\nu(c_I+0) = 1-\mu c_I = (2-\mu(4-\gamma\mu))/(2-2\mu)$, the full calculations of this section show it is in fact $f_\nu(c_I + 0) = 2-\mu\gamma/2$. This difference is a direct consequence of the correlations between real and imaginary parts of the self-energy, and that's why the solution proposed in [31] is fundamentally inapplicable to the RP models with heavy-tailed hopping distributions. At the end of the day, the solution (E.6) coincides with our numerical simulations up to the extrapolation precision, see Fig. 8, and the problem of analytical SFD calculation for Lévy-RP models can be considered solved.

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
