# Peer review of "Anatomy of the eigenstates distribution: a quest for a genuine multifractality"

_SciPost Physics, doi:SciPost Phys. 16, 008 (2024)_

## Round 1 · Referee Report · Anonymous (Referee 1) · 2023-11-10

Strengths

  • A nice graphical method is introduced.
  • Addresses a timely (and somehow controversial) issue, namely the possible existence of multifractal structure in generalized RP models.

Weaknesses

  • Better discussion with previous (contradicting ) results is needed
  • Presentation should be improved

Report

In this paper, the authors study the possible existence of multi-fractal phases
in generalized Rosenzweig-Porter (RP) models. They focus in particular in the cases
where the hopping terms (in the language of a related Anderson model) are
distributed according to (i) a Gaussian, (ii) a heavy/fat tailed and (iii) log-normal
distribution. By using a combination of a graphical computation method (which is
developed in Section 3 of the present paper) and the cavity method in its
diagonal approximation, they investigate the spectrum of fractal dimensions of the local density of states
in these different models. From these computations, they argue that these models can not host
any multi-fractal phases -- although previous works had actually claimed the contrary.
They further generalize this statement to a much wider class of RP models.

I believe that the present paper is an interesting contribution to the question of
multi-fractality, which is of great current interest in the context of many-body localisation.
The graphical method presented here is an interesting tool, which might be useful for further studies.
Besides, the arguments presented are interesting and physically sound -- though it
might not be the end of the story -- since most of these arguments are heuristic and
rely on a number of approximations but
do not constitute any mathematical proof (and the authors do not pretend for it).

My main criticism is that the authors should better discuss the discrepancy
between their results/statements and previous papers that had instead claimed the existence of
multifractal structures. In particular, in the context of the Lévy RP, where does the discrepancy
between their value of $D_\infty$ and the one computed in Ref. [29] ? Could they point out to
a mistake done in [29] ? Why should their approach be more reliable than the one in [29]? I think
a deeper discussion of that matter is needed. Besides, in many places in the paper, the reasoning and
rationale behind the arguments
could be explained better. Below is a list of more precise comments that the authors should consider
to clarify and improve the presentation of their results.

Requested changes

1) In the introduction, p.3, the following statement is a bit misleading:

"Fractal states are characterized by a single powerlaw tail in PDF which provides usually the only energy/time scale in the system in addition to the standard ones"

Usually, the existence of a power law behavior means that there is no typical energy/time scale.

2) A bit after that: what is meant, precisely, by "running exponent" ?

3) In the language of probability theory the algebra which is presented consists in manipulations of large deviation function. I would not be able to give a precise reference in the existing literature (in maths or physics) for that but maybe the authors should mention the analogy with large deviations and cite, for instance:

H Touchette, The large deviation approach to statistical mechanics, Physics Reports 478, 1 (2009)

4) Above Eq. (22): "This means that there can be nothing also to the left of the broadening typical value". It is not very clear what is meant by "broadening typical value".

5) Below Eq. (37), the authors write "where the right-wing tails are, as usual, eaten by the extensive number of the elements in the sum". What do the authors mean by "eaten" ? This does not sound as an appropriate word here !

6) The discussion below Fig. 7 is rather unclear and difficult to follow -- in particular the paragraph starting with "The reason why...". The authors should clarify a bit their reasoning.

7) On p. 16, below Fig. 7: "Indeed, providing the conditions above hold" --> "providing" should be "provided"

8) The results on Fig. 8 are very difficult to grasp since the presentation is very cumbersome and the caption does not contain very helpful informations. The authors should really make some effort to clarify its presentation/description (in the caption), since it contains important results for the Lévy RP model.

9) Below Eq. (48): "Formally speaking, this is already multifractality for $D_q$ with negative q": I guess "multifractality" should be "multifractal".

10) Again below Eq. (48): "From Eq. (48), one can find the part of the phase diagram, Fig. 10": which "part" are the authors talking about ?

11) In many place, "let's" should be replaced by "let us", similarly "haven't" should be "have not".

12) On p. 17: "providing" should be "provided"

  • validity: good
  • significance: high
  • originality: high
  • clarity: ok
  • formatting: good
  • grammar: reasonable

Author:  Anton Kutlin  on 2023-12-13  [id 4189]

(in reply to Report 1 on 2023-11-10)
Category:
answer to question

Dear Referee 1,

We are grateful for your careful reading of the manuscript and valuable comments.
Please see the attached reply, followed by the revised text with the changes highlighted in orange.

Sincerely yours,
the authors.

Attachment:

reply_1_orange_text_compressed.pdf

---

## Round 1 · Referee Report · Anonymous (Referee 2) · 2023-11-13

Strengths

1-cutting edge graphical method to compute the spectrum of fractal dimensions of the local density of states.
2-Concludes there is no multifractal phase in RP models

Weaknesses

1-restriction of results to RP models.

Report

The authors investigate the presence of non ergodic
extended (NEE) phases in random matrix models known as Rozensweig Porter (RP) (type) models. The different types of models depend on the tails of the distributions of the off-diagonal entries of these random matrices.

These phases are present in many systems at criticality like many-body systems undergoing many body localization transition, and also other simpler systems, but conjectured to be very related to the latter, like random regular graphs. The advantage of using random matrix models is that they provide the ability to arrive to analytical results that can provide deep insight into related more complicate problems. However, sometimes this advantage is also a disadvantage as their simplicity might restrict it’s application.

The main subject of this manuscript is then to study the existence of the NEE in RP models using a cutting-edge method graphical method to compute the spectrum of fractal dimensions SFD for the local density of states, which they also relate to the multifractal dimensions of the eigenfunctions. This method in itself justifies publication.

The authors provide meticulous calculations showing the validity of their method for different types of distribution tails, with different known properties, and arrive at the conclusion that in fact RP models without hooping term cannot host multifractal phase (only a fractal one).

This is indeed a nice result, although it can constrain generalizations of results of these types of models to others., unless hopping terms are added.

In any case, I find the manuscript is well written, very clearly explained (maybe a bit long), and relevant to the community that it is addressed to (which is wide). Therefor I recommend it to be accepted for publication.
  • validity: high
  • significance: good
  • originality: high
  • clarity: good
  • formatting: good
  • grammar: good

Author:  Anton Kutlin  on 2023-12-13  [id 4190]

(in reply to Report 2 on 2023-11-13)
Category:
answer to question

Dear Referee 2,

We are grateful for your careful reading of the manuscript and valuable comments.
Please see the attached reply, followed by the revised text with the changes highlighted in orange.

Sincerely yours,
the authors.

Attachment:

reply_2_orange_text_compressed.pdf

---

## Round 2 · Referee Report · Anonymous (Referee 3) · 2023-12-16

Report

In this revised version of the manuscript, the authors have carefully and satisfactorily taken into account me remarks and comments. Therefore I am glad to recommend the publication of the present manuscript in SciPost.

---

## Round 2 · List of Changes

Due to the request of Referee #1:

1. The explanation of the discrepancy between the results of the present paper and the paper [31] ([29] in the previous version) is added to the end of Sec.5
2. The notion of the running exponent and the meaning of the power law tails in PDFs of fractal distributions are clarified in the introduction.
3. The reference to "The large deviation approach to statistical mechanics" by H. Touchette is added, and the relation of our theory to the large deviation theory is discussed at the end of Sec.2
4. Above Eq. (22): the world order was changed to clarify the meaning of the "broadening typical value."
5. Below Eq. (37): the "eaten by the extensive number" slang was corrected.
6. Below Fig. 7, especially the paragraph starting from "The reason why...": the end of Sec. 5 was significantly rewritten to clarify several important points.
7. On p. 16, below Fig. 7, and in other places, the misprint was corrected, and "providing" was replaced with "provided."
8. Fig. 8 was updated, and its caption was extended to make it more accessible.
9. Below Eq. (48): "multifractality" was replaced with "multifractal."
10. We have extended the description of which part of the phase diagram we have meant after Eq. (48) and in the caption of Fig. 10.
11. We replaced "let's" with "let us" and "haven't" with "have not."

Due to the review from Referee #2, we have added a discussion of possible method generalizations to the conclusion.

---

## Editorial Decision

published